# Reinforcement Learning with General Value Function Approximation: Provably Efficient Approach via Bounded Eluder Dimension

**Ruosong Wang**
Carnegie Mellon University
ruosongw@andrew.cmu.edu

**Ruslan Salakhutdinov**
Carnegie Mellon University
rsalakhu@cs.cmu.edu

**Lin F. Yang**
University of California, Los Angeles
linyang@ee.ucla.edu

## Abstract

Value function approximation has demonstrated phenomenal empirical success in reinforcement learning (RL). Nevertheless, despite a handful of recent progress on developing theory for RL with linear function approximation, the understanding of *general* function approximation schemes largely remains missing. In this paper, we establish the first provably efficient RL algorithm with general value function approximation. We show that if the value functions admit an approximation with a function class $\mathcal{F}$, our algorithm achieves a regret bound of $\widetilde{O}(\mathrm{poly}(dH)\sqrt{T})$ where $d$ is a complexity measure of $\mathcal{F}$ that depends on the eluder dimension [Russo and Van Roy, 2013] and log-covering numbers, $H$ is the planning horizon, and $T$ is the number interactions with the environment. Our theory generalizes the linear MDP assumption to general function classes. Moreover, our algorithm is model-free and provides a framework to justify the effectiveness of algorithms used in practice.

## 1 Introduction

In reinforcement learning (RL), we study how an agent maximizes the cumulative reward by interacting with an unknown environment. RL finds enormous applications in a wide variety of domains, e.g., robotics [32], education [33], gaming-AI [50], etc. The unknown environment in RL is often modeled as a Markov decision process (MDP), in which there is a set of states $\mathcal{S}$ that describes all possible status of the environment. At a state $s \in \mathcal{S}$, an agent interacts with the environment by taking an action $a$ from an action space $\mathcal{A}$. The environment then transits to another state $s' \in \mathcal{S}$ which is drawn from some unknown transition distribution, and the agent also receives an immediate reward. The agent interacts with the environment episodically, where each episode consists of $H$ steps. The goal of the agent is to interact with the environment strategically such that after a certain number of interactions, sufficient information is collected so that the agent can act nearly optimally afterward. The performance of an agent is measured by the *regret*, which is defined as the difference between the total rewards collected by the agent and those a best possible agent would collect.

Without additional assumptions on the structure of the MDP, the best possible algorithm achieves a regret bound of $\widetilde{\Theta}(\sqrt{H|\mathcal{S}||\mathcal{A}|T})$[1] [7], where $T$ is the total number of steps the agent interacts with the environment. In other words, the algorithm learns to interact with the environment nearly as well

as an optimal agent after roughly $H|\mathcal{S}||\mathcal{A}|$ steps. This regret bound, however, can be unacceptably large in practice. E.g., the game of Go has a state space with size $3^{361}$, and the state space of certain robotics applications can even be continuous. Practitioners apply function approximation schemes to tackle this issue, i.e., the value of a state-action pair is approximated by a function which is able to predict the value of unseen state-action pairs given a few training samples. The most commonly used function approximators are deep neural networks (DNN) which have achieved remarkable success in playing video games [40], the game of Go [52], and controlling robots [3]. Nevertheless, despite the outstanding achievements in solving real-world problems, no convincing theoretical guarantees were known about RL with general value function approximators like DNNs.

Recently, there is a line of research trying to understand RL with simple function approximators, e.g. linear functions. For instance, given a feature extractor which maps state-action pairs to $d$-dimensional feature vectors, [63, 64, 29, 9, 41, 26, 67, 20, 61, 66, 19] developed algorithms with regret bound proportional to $\mathrm{poly}(dH)\sqrt{T}$ which is independent of the size of $\mathcal{S} \times \mathcal{A}$. Although being much more efficient than algorithms for the tabular setting, these algorithms require a well-designed feature extractor and also make restricted assumptions on the transition model. This severely limits the scope that these approaches can be applied to, since obtaining a good feature extractor is by no means easy and successful algorithms used in practice usually specify a function class (e.g. DNNs with a specific architecture) rather than a feature extractor. To our knowledge, the following fundamental question about RL with general function approximation remains unanswered at large:

*Does RL with general function approximation learn to interact with an unknown environment provably efficiently?*

In this paper, we address the above question by developing a provably efficient (both computationally and statistically) $Q$-learning algorithm that works with general value function approximators. To run the algorithm, we are only required to specify a value function class, *without* the need for feature extractors. Since this is the same requirement as algorithms used in practice like deep $Q$-learning [40], our theoretical guarantees on the algorithm provide a justification of why practical algorithms work so well. Furthermore, we show that our algorithm enjoys a regret bound of $\widetilde{O}(\mathrm{poly}(dH)\sqrt{T})$ where $d$ is a complexity measure of the function class that depends on the *eluder dimension* [48] and log-covering numbers. Our theory generalizes the linear MDP assumption in [63, 29] to general function classes, and our algorithm provides comparable regret bounds when applied to the linear case.

## 1.1 Related Work

**Tabular RL.** There is a long line of research on the sample complexity and regret bound for RL in the tabular setting. See, e.g., [31, 30, 55, 54, 25, 58, 6, 36, 14, 46, 47, 2, 7, 51, 15, 28, 65, 68, 60] and references therein. In particular, [25] proved a tight regret lower bound $\Omega(\sqrt{H|\mathcal{S}||\mathcal{A}|T})$ and [7] showed the first asymptotically tight regret upper bound $\widetilde{O}(\sqrt{H|\mathcal{S}||\mathcal{A}|T})$. Although these algorithms achieve asymptotically tight regret bounds, they can not be applied in problems with huge state space due to the linear dependency on $\sqrt{|\mathcal{S}|}$ in the regret bound. Moreover, the regret lower bound $\Omega(\sqrt{H|\mathcal{S}||\mathcal{A}|T})$ demonstrates that without further assumptions, RL with huge state space is information-theoretically hard to solve. In this paper, we exploit the structure that the value functions lie in a function class with bounded complexity and devise an algorithm whose regret bound scales polynomially in the complexity of the function class instead of the number of states.

**Bandits.** Another line of research studies bandits problems with linear function approximation [13, 1, 38]. These algorithms are later generalized to the generalized linear model [23, 37]. A novel work [48] studies bandits problems with general function approximation and proves that UCB-type algorithms and Thompson sampling achieve a regret bound of $\widetilde{O}(\sqrt{\dim_E \cdot \log(\mathcal{N})T})$ where $\dim_E$ is the *eluder dimension* of the function class and $\mathcal{N}$ is the covering number of the function class. In this paper we study the RL setting with general value function approximation, and the regret bound of our algorithm also depends on the eluder dimension and the log-covering number of the function class. However, we would like to stress that the RL setting is much more complicated than the bandits setting, since the bandits setting is a special case of the RL setting with planning horizon $H = 1$ and thus there is no state transition in the bandits setting.

**RL with Linear Function Approximation.** Recently there has been great interest in designing and analyzing algorithms for RL with linear function approximation. See, e.g., [63, 64, 29, 9, 41, 26, 67, 20, 61, 18, 66, 19]. These papers design provably efficient algorithms under the assumption that there is a well-designed feature extractor available to the agent and the value function or the model can be approximated by a linear function or a generalized linear function of the feature vectors. Moreover, the algorithm in [66] requires solving the Planning Optimization Program which could be computationally intractable. In this paper, we study RL with general function approximation in which case a feature extractor may not even be available, and our goal is to develop an efficient (both computationally and statistically) algorithm with provable regret bounds without making explicit assumptions on the model.

**RL with General Function Approximation.** It has been shown empirically that combining RL algorithms with neural network function approximators could lead to superior performance on various tasks [39, 49, 62, 59, 52, 3]. Theoretically, [45] analyzed the regret bound of Thompson sampling when applied to RL with general function approximation. Compared to our result, [45] makes explicit model-based assumptions (the transition operator and the reward function lie in a function class) and their regret bound depends on the global Lipschitz constant. In contrast, in this paper we focus on UCB-type algorithms with value-based assumptions, and our regret bound does not depend on the global Lipschitz constant. Recently, Ayoub et al. [5] proposed an algorithm for model-based RL with general function approximation based on value-targeted regression, and the regret bound of their algorithm also depends on the eluder dimension. On the contrary, in this paper we focus on value-based RL algorithms. In particular, for the case of linear functions, our assumption is equivalent to the linear MDP assumption in [63, 29], while the assumption in [5] is equivalent to the assumption that the true model is a linear combination of some known models [42, 69].

Recent theoretical progress has produced provably sample efficient algorithms for RL with general function approximation, but many of these algorithms are relatively impractical. In particular, [27, 56, 16] devised algorithms whose sample complexity or regret bound can be upper bounded in terms of the Bellman rank or the witness rank. However, these algorithms are not computationally efficient. The algorithm in [19] can also be applied in RL with general function approximation. However, their algorithms require the transition of the MDP to be deterministic. There is also a line of research analyzing Approximate Dynamic Programming (ADP) in RL with general function approximation [8, 43, 57, 4, 44, 10]. These papers focus on the batch RL setting, and there is no exploration components in the algorithms. The sample complexity of these algorithms usually depends on the concentrability coefficient and is thus incomparable to our results.

## 2 Preliminaries

Throughout the paper, for a positive integer $N$, we use $[N]$ to denote the set $\{1, 2, \ldots, N\}$.

**Episodic Markov Decision Process.** Let $M = (\mathcal{S}, \mathcal{A}, P, r, H, \mu)$ be a *Markov decision process* (MDP) where $\mathcal{S}$ is the state space, $\mathcal{A}$ is the action space with bounded size, $P : \mathcal{S} \times \mathcal{A} \to \Delta(\mathcal{S})$ is the transition operator which takes a state-action pair and returns a distribution over states, $r : \mathcal{S} \times \mathcal{A} \to [0, 1]$ is the deterministic reward function[2], $H \in \mathbb{Z}_+$ is the planning horizon (episode length), and $\mu \in \Delta(\mathcal{S})$ is the initial state distribution.

A policy $\pi$ chooses an action $a \in \mathcal{A}$ based on the current state $s \in \mathcal{S}$ and the time step $h \in [H]$. Formally, $\pi = \{\pi_h\}_{h=1}^H$ where for each $h \in [H]$, $\pi_h : \mathcal{S} \to \mathcal{A}$ maps a given state to an action. The policy $\pi$ induces a trajectory $s_1, a_1, r_1, s_2, a_2, r_2, \ldots, s_H, a_H, r_H$, where $s_1 \sim \mu$, $a_1 = \pi_1(s_1)$, $r_1 = r(s_1, a_1)$, $s_2 \sim P(s_1, a_1)$, $a_2 = \pi_2(s_2)$, etc.

An important concept in RL is the $Q$-function. Given a policy $\pi$, a level $h \in [H]$ and a state-action pair $(s, a) \in \mathcal{S} \times \mathcal{A}$, the $Q$-function is defined as $Q_h^\pi(s, a) = \mathbb{E}\left[\sum_{h'=h}^H r_{h'} \mid s_h = s, a_h = a, \pi\right]$. Similarly, the value function of a given state $s \in \mathcal{S}$ is defined as $V_h^\pi(s) = \mathbb{E}\left[\sum_{h'=h}^H r_{h'} \mid s_h = s, \pi\right]$.

We use $\pi^*$ to denote an optimal policy, i.e., $\pi^*$ is a policy that maximizes $\mathbb{E}\left[\sum_{h=1}^{H} r_h \mid \pi\right]$. We also denote $Q_h^*(s, a) = Q_h^{\pi^*}(s, a)$ and $V_h^*(s) = V_h^{\pi^*}(s)$.

In the episodic MDP setting, the agent aims to learn the optimal policy by interacting with the environment during a number of episodes. For each $k \in [K]$, at the beginning of the $k$-th episode, the agent chooses a policy $\pi^k$ which induces a trajectory, based on which the agent chooses policies for later episodes. We assume $K$ is fixed and known to the agent, though our algorithm and analysis can be readily generalized to the case that $K$ is unknown in advance. Throughout the paper, we define $T := KH$ to be the total number of steps that the agent interacts with the environment.

We adopt the following regret definition in this paper.

**Definition 1.** *The regret of an algorithm $\mathcal{A}$ after $K$ episodes is defined as* $\mathrm{Reg}(K) = \sum_{k=1}^{K} V_1^*\left(s_1^k\right) - V_1^{\pi^k}\left(s_1^k\right)$ *where $\pi^k$ is the policy played by algorithm $\mathcal{A}$ at the $k$-th episode.*

**Additional Notations.** For a function $f : \mathcal{S} \times \mathcal{A} \to \mathbb{R}$, define $\|f\|_\infty = \max_{(s,a) \in \mathcal{S} \times \mathcal{A}} |f(s, a)|$. Similarly, for a function $v : \mathcal{S} \to \mathbb{R}$, define $\|v\|_\infty = \max_{s \in \mathcal{S}} |v(s)|$. Given a dataset $\mathcal{D} = \{(s_i, a_i, q_i)\}_{i=1}^{|\mathcal{D}|} \subseteq \mathcal{S} \times \mathcal{A} \times \mathbb{R}$, for a function $f : \mathcal{S} \times \mathcal{A} \to \mathbb{R}$, define $\|f\|_\mathcal{D} = \left(\sum_{t=1}^{|\mathcal{D}|} (f(s_t, a_t) - q_t)^2\right)^{1/2}$. For a set of state-action pairs $\mathcal{Z} \subseteq \mathcal{S} \times \mathcal{A}$, for a function $f : \mathcal{S} \times \mathcal{A} \to \mathbb{R}$, define $\|f\|_\mathcal{Z} = \left(\sum_{(s,a) \in \mathcal{Z}} (f(s, a))^2\right)^{1/2}$. For a set of functions $\mathcal{F} \subseteq \{f : \mathcal{S} \times \mathcal{A} \to \mathbb{R}\}$, we define the width function of a state-action pair $(s, a)$ as $w(\mathcal{F}, s, a) = \max_{f, f' \in \mathcal{F}} f(s, a) - f'(s, a)$.

**Our Assumptions.** We make the following assumption on the $Q$-function throughout the paper.

**Assumption 1.** *There exists a set of functions $\mathcal{F} \subseteq \{f : \mathcal{S} \times \mathcal{A} \to [0, H + 1]\}$, such that for any $V : \mathcal{S} \to [0, H]$, there exists $f_V \in \mathcal{F}$ which satisfies*

$$f_V(s, a) = r(s, a) + \sum_{s' \in \mathcal{S}} P(s' \mid s, a) V(s') \quad \forall (s, a) \in \mathcal{S} \times \mathcal{A}. \tag{1}$$

Intuitively, Assumption 1 requires that for any $V : \mathcal{S} \to [0, H]$, after applying the Bellman backup operator, the resulting function lies in the function class $\mathcal{F}$. We note that Assumption 1 is very general and includes many previous assumptions as special cases. For instance, for the tabular RL setting, $\mathcal{F}$ can be the entire function space of $\mathcal{S} \times \mathcal{A} \to [0, H + 1]$. For linear MDPs [63, 64, 29, 61] where both the reward function $r : \mathcal{S} \times \mathcal{A} \to [0, 1]$ and the transition operator $P : \mathcal{S} \times \mathcal{A} \to \Delta(\mathcal{S})$ are linear functions of a given feature extractor $\phi : \mathcal{S} \times \mathcal{A} \to \mathbb{R}^d$, $\mathcal{F}$ can be defined as the class of linear functions with respect to $\phi$. In practice, when $\mathcal{F}$ is a function class with sufficient expressive power (e.g. deep neural networks), Assumption 1 (approximately) holds. In the supplementary material, we consider a misspecified setting where (1) only holds approximately, and we show that our algorithm still achieves provable regret bounds in the misspecified setting.

The complexity of $\mathcal{F}$ determines the learning complexity of the RL problem under consideration. To characterize the complexity of $\mathcal{F}$, we use the following definition of eluder dimension which was first introduced in [48] to characterize the complexity of different function classes in bandits problems.

**Definition 2** (Eluder dimension). *Let $\varepsilon \geq 0$ and $\mathcal{Z} = \{(s_i, a_i)\}_{i=1}^n \subseteq \mathcal{S} \times \mathcal{A}$ be a sequence of state-action pairs.*

- *A state-action pair $(s, a) \in \mathcal{S} \times \mathcal{A}$ is $\varepsilon$-dependent on $\mathcal{Z}$ with respect to $\mathcal{F}$ if any $f, f' \in \mathcal{F}$ satisfying $\|f - f'\|_\mathcal{Z} \leq \varepsilon$ also satisfies $|f(s, a) - f'(s, a)| \leq \varepsilon$.*

- *An $(s, a)$ is $\varepsilon$-independent of $\mathcal{Z}$ with respect to $\mathcal{F}$ if $(s, a)$ is not $\varepsilon$-dependent on $\mathcal{Z}$.*

- *The $\varepsilon$-eluder dimension $\dim_E(\mathcal{F}, \varepsilon)$ of a function class $\mathcal{F}$ is the length of the longest sequence of elements in $\mathcal{S} \times \mathcal{A}$ such that, for some $\varepsilon' \geq \varepsilon$, every element is $\varepsilon'$-independent of its predecessors.*

It has been shown in [48] that $\dim_E(\mathcal{F}, \varepsilon) \leq |\mathcal{S}||\mathcal{A}|$ when $\mathcal{S}$ and $\mathcal{A}$ are finite. When $\mathcal{F}$ is the class of linear functions, i.e., $f_\theta(s, a) = \theta^\top \phi(s, a)$ for a given feature extractor $\phi : \mathcal{S} \times \mathcal{A} \to \mathbb{R}^d$, $\dim_E(\mathcal{F}, \varepsilon) = O(d \log(1/\varepsilon))$. When $\mathcal{F}$ is the class generalized linear functions of the form

$f_\theta(s, a) = g(\theta^\top \phi(s, a))$ where $g$ is an increasing continuously differentiable function, $\dim_E(\mathcal{F}, \varepsilon) = O(dr^2 \log(\overline{h}/\varepsilon))$ where $r = \frac{\sup_{\theta,(s,a)\in\mathcal{S}\times\mathcal{A}} g'(\theta^\top \phi(s,a))}{\inf_{\theta,(s,a)\in\mathcal{S}\times\mathcal{A}} g'(\theta^\top \phi(s,a))}$ and $\overline{h} = \sup_{\theta,(s,a)\in\mathcal{S}\times\mathcal{A}} g'(\theta^\top \phi(s,a))$. In [45], it has been shown that when $\mathcal{F}$ is the class of quadratic functions, i.e., $f_\Lambda(s, a) = \phi(s, a)^\top \Lambda \phi(s, a)$ where $\Lambda \in \mathbb{R}^{d\times d}$, $\dim_E(\mathcal{F}, \varepsilon) = O(d^2 \log(1/\varepsilon))$.

We further assume the function class $\mathcal{F}$ and the state-action pairs $\mathcal{S} \times \mathcal{A}$ have bounded complexity in the following sense.

**Assumption 2.** *For any $\varepsilon > 0$, the following holds:*

1. *there exists an $\varepsilon$-cover $\mathcal{C}(\mathcal{F}, \varepsilon) \subseteq \mathcal{F}$ with size $|\mathcal{C}(\mathcal{F}, \varepsilon)| \leq \mathcal{N}(\mathcal{F}, \varepsilon)$, such that for any $f \in \mathcal{F}$, there exists $f' \in \mathcal{C}(\mathcal{F}, \varepsilon)$ with $\|f - f'\|_\infty \leq \varepsilon$;*

2. *there exists an $\varepsilon$-cover $\mathcal{C}(\mathcal{S}\times\mathcal{A}, \varepsilon)$ with size $|\mathcal{C}(\mathcal{S}\times\mathcal{A}, \varepsilon)| \leq \mathcal{N}(\mathcal{S}\times\mathcal{A}, \varepsilon)$, such that for any $(s, a) \in \mathcal{S} \times \mathcal{A}$, there exists $(s', a') \in \mathcal{C}(\mathcal{S}\times\mathcal{A}, \varepsilon)$ with $\max_{f\in\mathcal{F}} |f(s, a) - f(s', a')| \leq \varepsilon$.*

Assumption 2 requires both the function class $\mathcal{F}$ and the state-action pairs $\mathcal{S} \times \mathcal{A}$ have bounded covering numbers. Since our regret bound depends *logarithmically* on $\mathcal{N}(\mathcal{F}, \cdot)$ and $\mathcal{N}(\mathcal{S} \times \mathcal{A}, \cdot)$, it is acceptable for the covers to have exponential size. In particular, when $\mathcal{S}$ and $\mathcal{A}$ are finite, it is clear that $\log \mathcal{N}(\mathcal{F}, \varepsilon) = \widetilde{O}(|\mathcal{S}||\mathcal{A}|)$ and $\log \mathcal{N}(\mathcal{S} \times \mathcal{A}, \varepsilon) = \log(|\mathcal{S}||\mathcal{A}|)$. For the case of $d$-dimensional linear functions and generalized linear functions, $\log \mathcal{N}(\mathcal{F}, \varepsilon) = \widetilde{O}(d)$ and $\log \mathcal{N}(\mathcal{S} \times \mathcal{A}, \varepsilon) = \widetilde{O}(d)$. For quadratic functions, $\log \mathcal{N}(\mathcal{F}, \varepsilon) = \widetilde{O}(d^2)$ and $\log \mathcal{N}(\mathcal{S} \times \mathcal{A}, \varepsilon) = \widetilde{O}(d)$.

## 3 Algorithm

**Overview.** The full algorithm is formally presented in Algorithm 1. From a high-level point of view, our algorithm resembles least-square value iteration (LSVI) and falls in a similar framework as the algorithm in [29, 61]. At the beginning of each episode $k \in [K]$, we maintain a replay buffer $\{(s_h^\tau, a_h^\tau, r_h^\tau)\}_{(h,\tau)\in[H]\times[k-1]}$ which contains all existing samples. We set $Q_{H+1}^k = 0$, and calculate $Q_H^k, Q_{H-1}^k, \ldots, Q_1^k$ iteratively as follows. For each $h = H, H-1, \ldots, 1$,

$$f_h^k(\cdot, \cdot) \leftarrow \arg\min_{f\in\mathcal{F}} \sum_{\tau=1}^{k-1} \sum_{h'=1}^{H} \left( f(s_{h'}^\tau, a_{h'}^\tau) - \left( r_{h'}^\tau + \max_{a\in\mathcal{A}} Q_{h+1}^k(s_{h'+1}^\tau, a) \right) \right)^2 \qquad (2)$$

and define $Q_h^k(\cdot, \cdot) = \min\{f_h^k(\cdot, \cdot) + b_h^k(\cdot, \cdot), H\}$. Here, $b_h^k(\cdot, \cdot)$ is a bonus function to be defined shortly. The above equation optimizes a least squares objective to estimate the next step value. We then play the greedy policy with respect to $Q_h^k$ to collect data for the $k$-th episode. The above procedure is repeated until all the $K$ episodes are completed.

**Stable Upper-Confidence Bonus Function.** With more collected data, the least squares predictor is expected to return a better approximate the true $Q$-function. To encourage exploration, we carefully design a bonus function $b_h^k(\cdot, \cdot)$ which guarantees that, with high probability, $Q_{h+1}^k(s, a)$ is an overestimate of the one-step backup. The bonus function $b_h^k(\cdot, \cdot)$ is guaranteed to tightly characterize the estimation error of the one-step backup $r(\cdot, \cdot) + \sum_{s'\in\mathcal{S}} P(s' \mid \cdot, \cdot)V_{h+1}^k(s')$, where $V_{h+1}^k(\cdot) = \max_{a\in\mathcal{A}} Q_{h+1}^k(\cdot, a)$ is the value function of the next step. The bonus function $b_h^k(\cdot, \cdot)$ is designed by carefully prioritizing important data and hence is *stable* even when the replay buffer has large cardinality. A detailed explanation and implementation of $b_h^k(\cdot, \cdot)$ is provided in Section 3.1.

### 3.1 Stable UCB via Importance Sampling

In this section, we formally define the bonus function $b_h^k(\cdot, \cdot)$ used in Algorithm 1. The bonus function is designed to estimate the confidence interval of our estimate of the $Q$-function. In our algorithm, we define the bonus function to be the width function $b_h^k(\cdot, \cdot) = w(\mathcal{F}_h^k, \cdot, \cdot)$ where the confidence region $\mathcal{F}_h^k$ is defined so that $r(\cdot, \cdot) + \sum_{s'\in\mathcal{S}} P(s' \mid \cdot, \cdot)V_{h+1}^k(s') \in \mathcal{F}_h^k$ with high probability. By definition of the width function, $b_h^k(\cdot, \cdot)$ gives an upper bound on the confidence interval of the estimate of the $Q$-function, since the width function *maximizes* the difference between all pairs of $Q$-functions that lie in the confidence region. We note that similar ideas have been applied in the bandit literature [48],

**Algorithm 1** $\mathcal{F}$-LSVI($\delta$)

1: **Input**: failure probability $\delta \in (0,1)$ and number of episodes $K$
2: **for** episode $k = 1, 2, \ldots, K$ **do**
3:      Receive initial state $s_1^k \sim \mu$
4:      $Q_{H+1}^k(\cdot, \cdot) \leftarrow 0$ and $V_{H+1}^k(\cdot) \leftarrow 0$
5:      $\mathcal{Z}^k \leftarrow \{(s_{h'}^\tau, a_{h'}^\tau)\}_{(\tau, h') \in [k-1] \times [H]}$
6:      **for** $h = H, \ldots, 1$ **do**
7:          $\mathcal{D}_h^k \leftarrow \{(s_{h'}^\tau, a_{h'}^\tau, r_{h'}^\tau + V_{h+1}^k(s_{h'+1}^\tau, a))\}_{(\tau, h') \in [k-1] \times [H]}$
8:          $f_h^k \leftarrow \arg\min_{f \in \mathcal{F}} \|f\|_{\mathcal{D}_h^k}^2$
9:          $b_h^k(\cdot, \cdot) \leftarrow \texttt{Bonus}(\mathcal{F}, f_h^k, \mathcal{Z}^k, \delta)$ (Algorithm 3)
10:         $Q_h^k(\cdot, \cdot) \leftarrow \min\{f_h^k(\cdot, \cdot) + b_h^k(\cdot, \cdot), H\}$ and $V_h^k(\cdot) = \max_{a \in \mathcal{A}} Q_h^k(\cdot, a)$
11:         $\pi_h^k(\cdot) \leftarrow \arg\max_{a \in \mathcal{A}} Q_h^k(\cdot, a)$
12:      **for** $h = 1, 2, \ldots, H$ **do**
13:         Take action $a_h^k \leftarrow \pi_h^k(s_h^k)$ and observe $s_{h+1}^k \sim P(\cdot \mid s_h^k, a_h^k)$ and $r_h^k = r(s_h^k, a_h^k)$

in reinforcement learning with linear function approximation [20] and in reinforcement learning with general function apprximation in deterministic systems [19].

To define the confidence region $\mathcal{F}_h^k$, a natural definition would be $\mathcal{F}_h^k = \{f \in \mathcal{F} \mid \|f - f_h^k\|_{\mathcal{Z}^k}^2 \leq \beta\}$ where $\beta$ is defined so that $r(\cdot, \cdot) + \sum_{s' \in \mathcal{S}} P(s' \mid \cdot, \cdot) V_{h+1}^k(s') \in \mathcal{F}_h^k$ with high probability, and recall that $\mathcal{Z}^k = \{(s_{h'}^\tau, a_{h'}^\tau)\}_{(\tau, h') \in [k-1] \times [H]}$ is the set of state-action pairs defined in Line 5. However, as one can observe, the complexity of such a bonus function could be extremely high as it is defined by a dataset $\mathcal{Z}^k$ whose size can be as large as $T = KH$. A high-complexity bonus function could potentially introduce *instability* issues in the algorithm. Technically, we require a stable bonus function to allow for highly concentrated estimate of the one-step backup so that the confidence region $\mathcal{F}_h^k$ is accurate even for bounded $\beta$. Our strategy to "stabilize" the bonus function is to reduce the size of the dataset by importance sampling, so that only important state-action pairs are kept and those unimportant ones (which potentially induce instability) are ignored. Another benefit of reducing the size of the dataset is that it leads to superior computational complexity when evaluating the bonus function in practice. In later part of this section, we introduce an approach to estimate the importance of each state-action pair and a corresponding sampling method based on that. Finally, we note that importance sampling has also been applied in practical RL systems. For instance, in prioritized experience replay [49], the importance is measured by the TD error.

**Sensitivity Sampling.** Here we present a framework to subsample a given dataset, so that the confidence region is approximately preserved while the size of the dataset is greatly reduced. Our framework is built upon the *sensitivity sampling* technique introduced in a different context [35, 21, 22] to compress datasets. Our definition of sensitivity is similar to those in previous results [21, 22].

**Definition 3.** *For a given set of state-action pairs $\mathcal{Z} \subseteq \mathcal{S} \times \mathcal{A}$ and a function class $\mathcal{F}$, for each $z \in \mathcal{Z}$, define the $\lambda$-sensitivity of $(s, a)$ with respect to $\mathcal{Z}$ and $\mathcal{F}$ to be*

$$\mathsf{sensitivity}_{\mathcal{Z}, \mathcal{F}, \lambda}(s, a) = \max_{\substack{f, f' \in F \\ \|f - f'\|_{\mathcal{Z}}^2 \geq \lambda}} \frac{(f(s, a) - f'(s, a))^2}{\|f - f'\|_{\mathcal{Z}}^2}.$$

Sensitivity measures the importance of each data point $z$ in $\mathcal{Z}$ by considering the pair of functions $f, f' \in \mathcal{F}$ such that $z$ contributes the most to $\|f - f'\|_{\mathcal{Z}}^2$. In Algorithm 2, we define a procedure to sample each state-action pair with sampling probability proportional to the sensitivity. In this analysis, we show that after applying Algorithm 2 on the input dataset $\mathcal{Z}$, with high probability, the confidence region $\{f \in \mathcal{F} \mid \|f - f_h^k\|_{\mathcal{Z}}^2 \leq \beta\}$ is approximately preserved, while the size of the subsampled dataset is upper bounded by the eluder dimension of $\mathcal{F}$ times the log-covering number of $\mathcal{F}$.

**The Stable Bonus Function.** With the above sampling procedure, we are now ready to obtain a stable bonus function which is formally defined in Algorithm 3. In Algorithm 3, we first subsample the given dataset $\mathcal{Z}$ and then round the reference function $\bar{f}$ and all data points in the subsampled

---
**Algorithm 2** Sensitivity-Sampling($\mathcal{F}, \mathcal{Z}, \lambda, \varepsilon, \delta$)
---
1: **Input**: function class $\mathcal{F}$, set of state-action pairs $\mathcal{Z} \subseteq \mathcal{S} \times \mathcal{A}$, accuracy parameters $\lambda, \varepsilon > 0$ and failure probability $\delta \in (0, 1)$
2: Initialize $\mathcal{Z}' \leftarrow \{\}$
3: For each $z \in \mathcal{Z}$, let $p_z$ to be smallest real number such that $1/p_z$ is an integer and

$$p_z \geq \min\{1, \text{sensitivity}_{\mathcal{Z}, \mathcal{F}, \lambda}(z) \cdot 72 \ln(4\mathcal{N}(\mathcal{F}, \varepsilon/72 \cdot \sqrt{\lambda\delta/(|\mathcal{Z}|)})/\delta)/\varepsilon^2\} \qquad (3)$$

4: For each $z \in \mathcal{Z}$, independently add $1/p_z$ copies of $z$ into $\mathcal{Z}'$ with probability $p_z$
5: **return** $\mathcal{Z}'$
---

---
**Algorithm 3** Bonus($\mathcal{F}, \bar{f}, \mathcal{Z}, \delta$)
---
1: **Input**: function class $\mathcal{F}$, reference function $\bar{f} \in \mathcal{F}$, state-action pairs $\mathcal{Z} \subseteq \mathcal{S} \times \mathcal{A}$ and failure probability $\delta \in (0, 1)$
2: $\overline{\mathcal{Z}} \leftarrow$ Sensitivity-Sampling$\left(\mathcal{F}, \mathcal{Z}, \delta/(16T), 1/2, \delta\right)$          ▷ Subsample the dataset
3: $\overline{\mathcal{Z}} \leftarrow \{\}$ if $|\overline{\mathcal{Z}}| \geq 4T/\delta$ or the number of distinct elements in $\overline{\mathcal{Z}}$ exceeds

$$6912 \dim_E(\mathcal{F}, \delta/(16T^2)) \log(64H^2T^2/\delta) \ln T \ln(4\mathcal{N}(\mathcal{F}, \delta/(566T))/\delta)$$

4: Let $\widehat{f} \in \mathcal{C}(\mathcal{F}, 1/(8\sqrt{4T/\delta}))$ be such that $\|\bar{f} - \widehat{f}\|_\infty \leq 1/(8\sqrt{4T/\delta})$      ▷ Round $\bar{f}$
5: $\widehat{\mathcal{Z}} \leftarrow \{\}$
6: **for** $z \in \overline{\mathcal{Z}}$ **do**                               ▷ Round state-action pairs
7:      Let $\widehat{z} \in \mathcal{C}(\mathcal{S} \times \mathcal{A}, 1/(8\sqrt{4T/\delta}))$ be such that $\sup_{f \in \mathcal{F}} |f(z) - f(\widehat{z})| \leq 1/(8\sqrt{4T/\delta})$
8:      $\widehat{\mathcal{Z}} \leftarrow \widehat{\mathcal{Z}} \cup \{\widehat{z}\}$
9: **return** $\widehat{w}(\cdot, \cdot) := w(\widehat{\mathcal{F}}, \cdot, \cdot)$, where $\widehat{\mathcal{F}} = \left\{ f \in \mathcal{F} \mid \|f - \widehat{f}\|_{\widehat{\mathcal{Z}}}^2 \leq 3\beta(\mathcal{F}, \delta) + 2 \right\}$ and

$$\beta(\mathcal{F}, \delta) = c' H^2 \cdot \log^2(T/\delta) \cdot \dim_E(\mathcal{F}, \delta/T^3) \cdot \ln(\mathcal{N}(\mathcal{F}, \delta/T^2)/\delta) \cdot \log\left(\mathcal{N}(\mathcal{S} \times \mathcal{A}, \delta/T)\right) \cdot T/\delta \tag{4}$$

for some absolute constants $c' > 0$.
---

dataset $\overline{\mathcal{Z}}$ to their nearest neighbors in a $1/(8\sqrt{4T/\delta})$-cover. We discard the subsampled dataset if its size is too large (which happens with low probability as guaranteed by our analysis), and then define the confidence region using the new dataset and the rounded reference function.

We remark that in Algorithm 3, we round the reference function $\bar{f}$ and the state-action pairs in $\mathcal{Z}$ mainly for the purpose of theoretical analysis. In practice, the reference function and the state-action pairs are always stored with bounded precision, in which case explicit rounding is unnecessary. Moreover, when applying Algorithm 3 in practice, if the eluder dimension of the function class is unknown in advance, one may treat $\beta(\mathcal{F}, \delta)$ in (4) as a tunable parameter.

### 3.2 Computational Efficiency

Finally, we discuss how to implement our algorithm computationally efficiently. To implement Algorithm 1, in Line 8, one needs to solve an empirical risk minimization (ERM) problem which can often be efficiently solved using appropriate optimization methods. To implement Algorithm 3, one needs to evaluate the width function $w(\widehat{\mathcal{F}}, \cdot, \cdot)$ for a confidence region $\widehat{\mathcal{F}}$ of the form $\widehat{\mathcal{F}} = \left\{ f \in \mathcal{F} \mid \|f - \widehat{f}\|_{\mathcal{Z}}^2 \leq \beta \right\}$. To evaluate the width function, it suffices to have access to a regression oracle by invoking known reductions in [34, 24]. In particular, when $\mathcal{F}$ is the class of linear functions, there is a closed-form formula for the width function and thus the width function can be efficiently evaluated in this case. To implement Algorithm 2, one needs to efficiently estimate $\lambda$-sensitivity of all state-action pairs in a given set $\mathcal{Z}$. When $\mathcal{F}$ is the class of linear functions, sensitivity is equivalent to leverage score [17] which can be efficiently estimated [12, 11]. For a general function class $\mathcal{F}$, we give an algorithm to estimate the $\lambda$-sensitivity in the supplementary material which is computationally efficient if one can efficiently judge whether a given state-action pair $z$ is $\varepsilon$-independent of a set of state-actions pairs $\mathcal{Z}$ with respect to $\mathcal{F}$ for a given $\varepsilon > 0$. Note that judging whether a given

state-action pair is $\varepsilon$-independent of a set of state-action pairs can be again reduced to evaluating the width function.

We believe that the running time of our algorithm can be further reduced by using the doubling trick or online sampling algorithms, and we leave it as a future work to further optimize the running time.

## 4 Theoretical Guarantee

In this section we provide the theoretical guarantee of Algorithm 1, which is stated in Theorem 1.

**Theorem 1.** *Under Assumption 1, after interacting with the environment for $T = KH$ steps, with probability $1 - \delta$, Algorithm 1 achieves a a regret bound of $\mathrm{Reg}(K) \leq \sqrt{\iota \cdot H^2 \cdot T}$, where*

$$\iota \leq C \cdot \log^2 (T/\delta) \cdot \dim_E^2 \left( \mathcal{F}, \delta/T^3 \right) \cdot \ln \left( \mathcal{N} \left( \mathcal{F}, \delta/T^2 \right)/\delta \right) \cdot \log \left( \mathcal{N} \left( \mathcal{S} \times \mathcal{A}, \delta/T \right) \cdot T/\delta \right)$$

*for some constant $C > 0$.*

**Remark 1.** *For the tabular setting, we may set $\mathcal{F}$ to be the entire function space of $\mathcal{S} \times \mathcal{A} \to [0, H+1]$. Recall that when $\mathcal{S}$ and $\mathcal{A}$ are finite, for any $\varepsilon > 0$, $\dim_E(\mathcal{F}, \varepsilon) \leq |\mathcal{S}||\mathcal{A}|$, $\log(\mathcal{N}(\mathcal{F}, \varepsilon)) = \widetilde{O}(|\mathcal{S}||\mathcal{A}|)$ and $\log(\mathcal{N}(\mathcal{S} \times \mathcal{A}, \varepsilon)) = O(\log(|\mathcal{S}||\mathcal{A}|))$, and thus the regret bound in Theorem 1 is $\widetilde{O}(\sqrt{|\mathcal{S}|^3|\mathcal{A}|^3 H^2 T})$ which is worse than the near-optimal bound in [7]. By a more refined analysis specialized to the tabular setting, the regret bound of our algorithm can be improved to $\widetilde{O}(\sqrt{|\mathcal{S}|^2|\mathcal{A}|^2 H^2 T})$. We would like to stress that our algorithm and analysis tackle a much more general setting and recovering the optimal regret bound for the tabular setting is not the focus of the current paper.*

**Remark 2.** *When $\mathcal{F}$ is the class of $d$-dimensional linear functions, we have $\dim_E(\mathcal{F}, \varepsilon) = \widetilde{O}(d)$, $\log(\mathcal{N}(\mathcal{F}, \varepsilon)) = \widetilde{O}(d)$ and $\log(\mathcal{N}(\mathcal{S} \times \mathcal{A}, \varepsilon)) = \widetilde{O}(d)$ and thus the regret bound in Theorem 1 is $\widetilde{O}(\sqrt{d^4 H^2 T})$, which is worse by a $\widetilde{O}(\sqrt{d})$ factor when compared to the bound in [29, 61], and is worse by a $\widetilde{O}(d)$ factor when compared to the bound in [66]. Note that for our algorithm, a regret bound of $\widetilde{O}(\sqrt{d^3 H^2 T})$ is achievable using a more refined analysis (see Remark 3) for the linear case which matches the results in [29, 61]. Moreover, unlike our algorithm, the algorithm in [66] requires solving the Planning Optimization Program and is thus computationally intractable. Finally, we would like to stress that our algorithm and analysis tackle the case that $\mathcal{F}$ is a general function class which contains the linear case studied in [29, 61] as a special case.*

Here we provide an overview of the proof to highlight the technical novelties in the analysis. The formal proof is provided in the supplementary material.

**The Stable Bonus Function.** Similar to the analysis in [29, 61], to account for the dependency structure in the data sequence, we need to bound the complexity of the bonus function $b_h^k(\cdot, \cdot)$. When $\mathcal{F}$ is the class of $d$-dimensional linear functions (as in [29, 61]), $b(\cdot, \cdot) = \|\phi(\cdot, \cdot)\|_{\Lambda^{-1}}$ for a covariance matrix $\Lambda \in \mathbb{R}^{d \times d}$, whose complexity is upper bounded by $d^2$ which is the number of entries in the covariance matrix $\Lambda$. However, such simple complexity upper bound is no longer available for the class of general functions considered in this paper. Instead, we bound the complexity of the bonus function by relying on the fact that the subsampled dataset has bounded size. Scrutinizing the sampling algorithm (Algorithm 2), it can be seen that the size of the subsampled dataset is upper bounded by the sum of the sensitivity of the data points in the given dataset times the log-convering number of the function class $\mathcal{F}$. To upper bound the sum of the sensitivity of the data points in the given dataset, we rely on a novel combinatorial argument which establishes a surprising connection between the sum of the sensitivity and the eluder dimension of the function class $\mathcal{F}$. We show that the sum of the sensitivity of data points is upper bounded by the eluder dimension of the dataset up to logarithm factors. Hence, the complexity of the subsampled dataset, and therefore, the complexity of the bonus function, is upper bound by the log-covering number of $\mathcal{S} \times \mathcal{A}$ (the complexity of each state-action pair) times the product of the eluder dimension of the function class and the log-covering number of the function class (the number of data points in the subsampled dataset).

In order to show that the confidence region is approximately preserved when using the subsampled dataset $\mathcal{Z}'$, we show that for any $f, f' \in \mathcal{F}$, $\|f - f'\|_{\mathcal{Z}'}^2$ is a good approximation to $\|f - f'\|_{\mathcal{Z}}^2$. To show this, we apply a union bound over all pairs of functions on the cover of $\mathcal{F}$ which allows us to consider fixed $f, f' \in \mathcal{F}$. For fixed $f, f' \in \mathcal{F}$, note that $\|f - f'\|_{\mathcal{Z}'}^2$ is an unbiased estimate of

$\|f - f'\|^2_{\mathcal{Z}}$, and importance sampling proportinal to the sensitivity implies an upper bound on the variance of the estimator which allows us to apply concentration bounds to prove the desired result. We note that the sensitivity sampling framework used here is very crucial to the theoreical guarantee of the algorithm. If one replaces sensitivity sampling with more naïve sampling approaches (e.g. uniform sampling), then the required sampling size would be much larger, which does not give any meaningful reduction on the size of the dataset and also leads to a high complexity bonus function.

**Remark 3.** *When $\mathcal{F}$ is the class of $d$-dimensional linear functions, our upper bound on the size of the subsampled dataset is $\widetilde{O}(d^2)$. However, in this case, our sampling algorithm (Algorithm 2) is equivalent to the leverage score sampling [17] and therefore the sample complexity can be further improved to $\widetilde{O}(d)$ using a more refined analysis [53]. Therefore, our regret bound can be improved to $\widetilde{O}(\sqrt{d^3 H^2 T})$, which matches the bounds in [29, 61]. However, the $\widetilde{O}(d)$ sample bound is specialized to the linear case and heavily relies on the matrix Chernoff bound which is unavailable for the class of general functions considered in this paper. This also explains why our regret bound in Theorem 1, when applied to the linear case, is larger by a $\sqrt{d}$ factor when compared to those in [29, 61]. We leave it as an open question to obtain more refined bound on the size of the subsampled dataset and improve the overall regret bound of our algorithm.*

**The Confidence Region.** Our algorithm applies the principle of optimism in the face of uncertainty (OFU) to balance exploration and exploitation. Note that $V^k_{h+1}$ is the value function estimated at step $h + 1$. In our analysis, we require the $Q$-function $Q^k_h$ estimated at level $h$ to satisfy $Q^k_h(\cdot, \cdot) \geq r(\cdot, \cdot) + \sum_{s' \in \mathcal{S}} P(s'|\cdot, \cdot) V^k_{h+1}(s')$ with high probability. To achieve this, we optimize the least squares objective to find a solution $f^k_h \in \mathcal{F}$ using collected data. We then show that $f^k_h$ is close to $r(\cdot, \cdot) + \sum_{s' \in \mathcal{S}} P(s'|\cdot, \cdot) V^k_{h+1}(s')$. This would follow from standard analysis if the collected samples were independent of $V^k_{h+1}$. However, $V^k_{h+1}$ is calculated using the collected samples and thus they are subtly dependent on each other. To tackle this issue, we notice that $V^k_{h+1}$ is computed by using $f^k_{h+1}$ and the bonus function $b^k_{h+1}$, and both $f^k_{h+1}$ and the bonus function $b^k_{h+1}$ have bounded complexity, thanks to the design of bonus function. Hence, we can construct a $1/T$-cover to approximate $V^k_{h+1}$. By doing so, we can now bound the fitting error of $f^k_h$ by replacing $V^k_{h+1}$ with its closest neighbor in the $1/T$-cover which is independent of the dataset. By a union bound over all functions in the $1/T$-cover, it follows that with high probability, $r(\cdot, \cdot) + \sum_{s' \in \mathcal{S}} P(s'|\cdot, \cdot) V^k_{h+1}(s') \in \left\{ f \in \mathcal{F} \mid \|f - f^k_h\|^2_{\mathcal{Z}^k} \leq \beta \right\}$ for some $\beta$ that depends only on the complexity of the bonus function and the function class $\mathcal{F}$.

**Regret Decomposition and the Eluder Dimension.** By standard regret decomposition for optimistic algorithms, the total regret is upper bounded by the summation of the bonus function $\sum_{k=1}^K \sum_{h=1}^H b^k_h \left( s^k_h, a^k_h \right)$. To bound the summation of the bonus function, we use an argument similar to that in [48], which shows that the summation of the bonus function can be upper bounded in terms of the eluder dimension of the function class $\mathcal{F}$, if the confidence region is defined using the original dataset. In the formal analysis (Lemma 9 and Lemma 10), we adapt the argument in [48] (more specifically, Proposition 3 and Lemma 2 in [48]) to show that even if the confidence region is defined using the subsampled dataset, the summation of the bonus function can be bounded in a similar manner.

# 5 Conclusion

In this paper, we give the first provably efficient value-based RL algorithm with general function approximation. Our algorithm achieves a regret bound of $\widetilde{O}(\text{poly}(dH)\sqrt{T})$ where $d$ is a complexity measure that depends on the eluder dimension and log-covering numbers of the function class. One interesting future direction is to extend our results to policy-based methods, by combining our techniques with, e.g., those in [9].

## Broader Impact

This work is mainly theoretical. By devising a provably efficient RL algorithm with general value function approximation, we believe our various theoretical insights could potentially guide practitioners to build theoretically-principled and robust RL systems.

## Disclosure of Funding

Ruosong Wang and Ruslan Salakhutdinov were supported in part by NSF IIS1763562, US Army W911NF1920104 and ONR Grant N000141812861.

## Footnotes

[1]Throughout the paper, we use $\widetilde{O}(\cdot)$ to suppress logarithmic factors.

[2]We assume the reward function is deterministic only for notational convience. Our results can be readily generalized to the case that rewards are stochastic.

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
