[Supplementary Material]

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

_{h+1}^k$ is computed by using $f_{h+1}^k$ and the bonus function $b_{h+1}^k$, and both $f_{h+1}^k$ and the bonus function $b_{h+1}^k$ have bounded complexity, thanks to the design of bonus function. Hence, we can construct a $1/T$-cover to approximate $V_{h+1}^k$. By doing so, we can now bound the fitting error of $f_h^k$ by replacing $V_{h+1}^k$ with its closest neighbor in the $1/T$-cover which is independent of the dataset. By a union bound over all functions in the $1/T$-cover, it follows that with high probability, $r(\cdot, \cdot) + \sum_{s' \in \mathcal{S}} P(s'|\cdot, \cdot) V_{h+1}^k(s') \in \left\{ f \in \mathcal{F} \mid \|f - f_h^k\|_{\mathcal{Z}^k}^2 \leq \beta \right\}$ for some $\beta$ that depends only on the complexity of the bonus function and the function class $\mathcal{F}$.

**Regret Decomposition and the Eluder Dimension.** By standard regret decomposition for optimistic algorithms, the total regret is upper bounded by the summation of the bonus function $\sum_{k=1}^K \sum_{h=1}^H b_h^k \left( s_h^k, a_h^k \right)$. To bound the summation of the bonus function, we use an argument similar to that in [48], which shows that the summation of the bonus function can be upper bounded in terms of the eluder dimension of the function class $\mathcal{F}$, if the confidence region is defined using the original dataset. In the formal analysis (Lemma 9 and Lemma 10), we adapt the argument in [48] (more specifically, Proposition 3 and Lemma 2 in [48]) to show that even if the confidence region is defined using the subsampled dataset, the summation of the bonus function can be bounded in a similar manner.

## 5   Conclusion

In this paper, we give the first provably efficient value-based RL algorithm with general function approximation. Our algorithm achieves a regret bound of $\widetilde{O}(\mathrm{poly}(dH)\sqrt{T})$ where $d$ is a complexity measure that depends on the eluder dimension and log-covering numbers of the function class. One interesting future direction is to extend our results to policy-based methods, by combining our techniques with, e.g., those in [9].

## Broader Impact

This work is mainly theoretical. By devising a provably efficient RL algorithm with general value function approximation, we believe our various theoretical insights could potentially guide practitioners to build theoretically-principled and robust RL systems.

## Disclosure of Funding

Ruosong Wang and Ruslan Salakhutdinov were supported in part by NSF IIS1763562, US Army W911NF1920104 and ONR Grant N000141812861.

## Footnotes

[1] Throughout the paper, we use $\widetilde{O}(\cdot)$ to suppress logarithmic factors.

[2]We assume the reward function is deterministic only for notational convience. Our results can be readily generalized to the case that rewards are stochastic.

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

# A  Missing Proofs

## A.1  Analysis of the Stable Bonus Function

Our first lemma gives an upper bound on the sum of the sensitivity in terms of the eluder dimension of the function class $\mathcal{F}$.

**Lemma 1.** *For a given set of state-action pairs $\mathcal{Z}$,*

$$\sum_{z \in \mathcal{Z}} \text{sensitivity}_{\mathcal{Z},\mathcal{F},\lambda}(z) \le 4\text{dim}_E(\mathcal{F}, \lambda/|\mathcal{Z}|)\log((H+1)^2|\mathcal{Z}|/\lambda)\ln|\mathcal{Z}|.$$

*Proof.* For each $z \in \mathcal{Z}$, let $f, f' \in F$ be an arbitrary pair of functions such that $\|f - f'\|_{\mathcal{Z}}^2 \ge \lambda$ and

$$\frac{(f(z) - f'(z))^2}{\|f - f'\|_{\mathcal{Z}}^2}$$

is maximized, and we define $L(z) = (f(z) - f'(z))^2$ for such $f$ and $f'$. Note that $0 \le L(z) \le (H+1)^2$. Let $\mathcal{Z} = \bigcup_{\alpha=0}^{\log((H+1)^2|\mathcal{Z}|/\lambda)-1} \mathcal{Z}^\alpha \cup \mathcal{Z}^\infty$ be a dyadic decomposition with respect to $L(\cdot)$, where for each $0 \le \alpha < \log((H+1)^2|\mathcal{Z}|/\lambda)$, define

$$\mathcal{Z}^\alpha = \{z \in \mathcal{Z} \mid L(z) \in ((H+1)^2 \cdot 2^{-\alpha-1}, (H+1)^2 \cdot 2^{-\alpha}]\}$$

and

$$\mathcal{Z}^\infty = \{z \in \mathcal{Z} \mid L(z) \le \lambda/|\mathcal{Z}|\}.$$

Clearly, for any $z \in \mathcal{Z}^\infty$, $\text{sensitivity}_{\mathcal{Z},\mathcal{F},\lambda}(z) \le 1/|\mathcal{Z}|$ and thus

$$\sum_{z \in \mathcal{Z}^\infty} \text{sensitivity}_{\mathcal{Z},\mathcal{F},\lambda}(z) \le 1.$$

Now we bound $\sum_{z \in \mathcal{Z}^\alpha} \text{sensitivity}_{\mathcal{Z},\mathcal{F},\lambda}(z)$ for each $0 \le \alpha < \log((H+1)^2|\mathcal{Z}|/\lambda)$ separately. For each $\alpha$, let $N_\alpha = |\mathcal{Z}^\alpha|/\text{dim}_E(\mathcal{F}, (H+1)^2 \cdot 2^{-\alpha-1})$ and we decompose $\mathcal{Z}^\alpha$ into $N_\alpha + 1$ disjoint subsets, i.e., $\mathcal{Z}^\alpha = \bigcup_{j=1}^{N_\alpha+1} \mathcal{Z}_j^\alpha$, by using the following procedure. Let $\mathcal{Z}^\alpha = \{z_1, z_2, \ldots, z_{|\mathcal{Z}^\alpha|}\}$ and we consider each $z_i$ sequentially. Initially $\mathcal{Z}_j^\alpha = \{\}$ for all $j$. Then, for each $z_i$, we find the largest $1 \le j \le N_\alpha$ such that $z_i$ is $(H+1)^2 \cdot 2^{-\alpha-1}$-independent of $\mathcal{Z}_j^\alpha$ with respect to $\mathcal{F}$. We set $j = N_\alpha + 1$ if such $j$ does not exist, and use $j(z_i) \in [N_\alpha + 1]$ to denote the choice of $j$ for $z_i$. If $j \le N_\alpha$ then we add $z_i$ to $\mathcal{Z}_j^\alpha$. By the design of the algorithm, for each $z_i$, it is clear that $z_i$ is dependent on each of $\mathcal{Z}_1^\alpha, \mathcal{Z}_2^\alpha, \ldots, \mathcal{Z}_{j(z_i)-1}^\alpha$.

Now we show that for each $z_i \in \mathcal{Z}^\alpha$,

$$\text{sensitivity}_{\mathcal{Z},\mathcal{F},\lambda}(z_i) \le 2/j(z_i).$$

For any $z_i \in \mathcal{Z}^\alpha$, we use $f, f' \in F$ to denote the pair of functions in $\mathcal{F}$ such that $\|f - f'\|_{\mathcal{Z}}^2 \ge \lambda$ and

$$\frac{(f(z_i) - f'(z_i))^2}{\|f - f'\|_{\mathcal{Z}}^2}$$

is maximized. Since $z_i \in \mathcal{Z}^\alpha$, we must have $(f(z_i) - f'(z_i))^2 > (H+1)^2 \cdot 2^{-\alpha-1}$. Since $z_i$ is dependent on each of $\mathcal{Z}_1^\alpha, \mathcal{Z}_2^\alpha, \ldots, \mathcal{Z}_{j(z_i)-1}^\alpha$, for each $1 \le k < j(z_i)$, we have

$$\|f - f'\|_{\mathcal{Z}_k^\alpha} \ge (H+1)^2 \cdot 2^{-\alpha-1},$$

which implies

$$\text{sensitivity}_{\mathcal{Z},\mathcal{F},\lambda}(z_i) = \frac{(f(z_i) - f'(z_i))^2}{\|f - f'\|_{\mathcal{Z}}^2} \le \frac{(H+1)^2 \cdot 2^{-\alpha}}{\|f - f'\|_{\mathcal{Z}}^2}$$

$$\le \frac{(H+1)^2 \cdot 2^{-\alpha}}{\sum_{k=1}^{j(z_i)-1} \|f - f'\|_{\mathcal{Z}_k^\alpha} + (f(z_i) - f'(z_i))^2} \le 2/j(z_i).$$

Moreover, by the definition of $(H+1)^2 \cdot 2^{-\alpha-1}$-independence, we have $|\mathcal{Z}_j^\alpha| \leq \dim_E(\mathcal{F}, (H+1)^2 \cdot 2^{-\alpha-1})$ for all $1 \leq j \leq N_\alpha$. Therefore,

$$\sum_{z \in \mathcal{Z}^\alpha} \mathsf{sensitivity}_{\mathcal{Z},\mathcal{F},\lambda}(z) \leq \sum_{1 \leq j \leq N_\alpha} |\mathcal{Z}_j^\alpha| \cdot 2/j + \sum_{z \in \mathcal{Z}_{N_\alpha+1}^\alpha} 2/N_\alpha$$

$$\leq 2\dim_E(\mathcal{F}, (H+1)^2 \cdot 2^{-\alpha-1}) \ln(N_\alpha) + |\mathcal{Z}^\alpha| \cdot \frac{2\dim_E(\mathcal{F}, (H+1)^2 \cdot 2^{-\alpha-1})}{|\mathcal{Z}^\alpha|}$$

$$\leq 3\dim_E(\mathcal{F}, (H+1)^2 \cdot 2^{-\alpha-1}) \ln(|\mathcal{Z}|).$$

By the monotonicity of eluder dimension, it follows that

$$\sum_{z \in \mathcal{Z}} \mathsf{sensitivity}_{\mathcal{Z},\mathcal{F},\lambda}(z)$$

$$\leq \sum_{\alpha=0}^{\log((H+1)^2|\mathcal{Z}|/\lambda)-1} \sum_{z \in \mathcal{Z}^\alpha} \mathsf{sensitivity}_{\mathcal{Z},\mathcal{F},\lambda}(z) + \sum_{z \in \mathcal{Z}^\infty} \mathsf{sensitivity}_{\mathcal{Z},\mathcal{F},\lambda}(z)$$

$$\leq 3\log((H+1)^2|\mathcal{Z}|/\lambda)\dim_E(\mathcal{F}, \lambda/|\mathcal{Z}|) \ln(|\mathcal{Z}|) + 1$$

$$\leq 4\log((H+1)^2|\mathcal{Z}|/\lambda)\dim_E(\mathcal{F}, \lambda/|\mathcal{Z}|) \ln(|\mathcal{Z}|).$$

$\square$

Using Lemma 1, we can prove an upper bound on the number of distinct elements in $\mathcal{Z}'$ returned by the sampling algorithm (Algorithm 2).

**Lemma 2.** *With probability at least* $1 - \delta/4$*, the number of distinct elements in* $\mathcal{Z}'$ *returned by Algorithm 2 is at most*

$$1728\dim_E(\mathcal{F}, \lambda/|\mathcal{Z}|) \log((H+1)^2|\mathcal{Z}|/\lambda) \ln(|\mathcal{Z}|) \ln(4\mathcal{N}(\mathcal{F}, \varepsilon/72 \cdot \sqrt{\lambda\delta/(|\mathcal{Z}|)})/\delta)/\varepsilon^2.$$

*Proof.* Note that $p_z \leq \min\{1, 2 \cdot \mathsf{sensitivity}_{\mathcal{Z},\mathcal{F},\lambda}(z) \cdot 72\ln(4\mathcal{N}(\mathcal{F}, \varepsilon/72 \cdot \sqrt{\lambda\delta/(|\mathcal{Z}|)})/\delta)/\varepsilon^2\}$, since for any real number $x < 1$, there always exists $\widehat{x} \in [x, 2x]$ such that $1/\widehat{x}$ is an integer. Let $X_z$ be a random variable defined as

$$X_z = \begin{cases} 1 & z \in Z' \\ 0 & z \notin Z' \end{cases}.$$

Clearly, the number of distinct elements in $\mathcal{Z}'$ is upper bounded by $\sum_{z \in \mathcal{Z}} X_z$ and $\mathbb{E}[X_z] = p_z$. By Lemma 1,

$$\sum_{z \in \mathcal{Z}} \mathbb{E}[X_z] \leq 576\dim_E(\mathcal{F}, \lambda/|\mathcal{Z}|) \log((H+1)^2|\mathcal{Z}|/\lambda) \ln(|\mathcal{Z}|) \ln(4\mathcal{N}(\mathcal{F}, \varepsilon/72 \cdot \sqrt{\lambda\delta/(|\mathcal{Z}|)})/\delta)/\varepsilon^2.$$

By Chernoff bound, with probability at least $1 - \delta/4$, we have

$$\sum_{z \in \mathcal{Z}} X_z \geq 1728\dim_E(\mathcal{F}, \lambda/|\mathcal{Z}|) \log((H+1)^2|\mathcal{Z}|/\lambda) \ln(|\mathcal{Z}|) \ln(4\mathcal{N}(\mathcal{F}, \varepsilon/72 \cdot \sqrt{\lambda\delta/(|\mathcal{Z}|)})/\delta)/\varepsilon^2.$$

$\square$

Our second lemma upper bounds the number of elements in $\mathcal{Z}'$ returned by Algorithm 2.

**Lemma 3.** *With probability at least* $1 - \delta/4$*,* $|\mathcal{Z}'| \leq 4|\mathcal{Z}|/\delta$*.*

*Proof.* Let $X_z$ be the random variable which is defined as

$$X_z = \begin{cases} 1/p_z & z \text{ is added into } \mathcal{Z}' \\ 0 & \text{otherwise} \end{cases}.$$

Note that $|\mathcal{Z}'| = \sum_{z \in \mathcal{Z}} X_z$ and $\mathbb{E}[X_z] = 1$. By Markov inequality, with probability $1 - \delta/4$, $|\mathcal{Z}'| \leq 4|\mathcal{Z}|/\delta$.

$\square$

Our third lemma shows that for the given set of state-action pairs $\mathcal{Z}$ and function class $\mathcal{F}$, Algorithm 2 returns a set of state-action pairs $\mathcal{Z}'$ so that $\|f - f'\|_{\mathcal{Z}}^2$ is approximately preserved for all $f, f' \in \mathcal{F}$.

**Lemma 4.** *With probability at least $1 - \delta/2$, for any $f, f' \in \mathcal{F}$,*

$$(1 - \varepsilon)\|f - f'\|_{\mathcal{Z}}^2 - 2\lambda \leq \|f - f'\|_{\mathcal{Z}'}^2 \leq (1 + \varepsilon)\|f - f'\|_{\mathcal{Z}}^2 + 8|\mathcal{Z}|\lambda/\delta.$$

*Proof.* In our proof, we separately consider two cases: $\|f - f'\|_{\mathcal{Z}}^2 < 2\lambda$ and $\|f - f'\|_{\mathcal{Z}}^2 \geq 2\lambda$.

**Case I: $\|f - f'\|_{\mathcal{Z}}^2 < 2\lambda$.** Consider $f, f' \in \mathcal{F}$ with $\|f - f'\|_{\mathcal{Z}}^2 < 2\lambda$. Conditioned on the event defined in Lemma 3 which holds with probability at least $1 - \delta/4$, we have $\|f - f'\|_{\mathcal{Z}'}^2 \leq |\mathcal{Z}'| \cdot \|f - f'\|_{\mathcal{Z}}^2 \leq 8|\mathcal{Z}|\lambda/\delta$. Moreover, we always have $\|f - f'\|_{\mathcal{Z}'} \geq 0$. In summary, we have

$$\|f - f'\|_{\mathcal{Z}}^2 - 2\lambda \leq \|f - f'\|_{\mathcal{Z}'}^2 \leq \|f - f'\|_{\mathcal{Z}}^2 + 8|\mathcal{Z}|\lambda/\delta.$$

**Case II: $\|f - f'\|_{\mathcal{Z}}^2 \geq 2\lambda$.** We first show that for any fixed $f, f' \in \mathcal{F}$ with $\|f - f'\|_{\mathcal{Z}}^2 \geq \lambda$, with probability at least $1 - \delta/(4\mathcal{N}(\mathcal{F}, \varepsilon/72 \cdot \sqrt{\lambda\delta/(|\mathcal{Z}|)}))$, we have

$$(1 - \varepsilon/4)\|f - f'\|_{\mathcal{Z}}^2 \leq \|f - f'\|_{\mathcal{Z}'}^2 \leq (1 + \varepsilon/4)\|f - f'\|_{\mathcal{Z}}^2.$$

To prove this, for each $z \in \mathcal{Z}$, define

$$X_z = \begin{cases} \frac{1}{p_z}(f(z) - f'(z))^2 & z \text{ is added into } \mathcal{Z}' \text{ for } 1/p_z \text{ times} \\ 0 & \text{otherwise} \end{cases}.$$

Clearly, $\|f - f'\|_{\mathcal{Z}'} = \sum_{z \in \mathcal{Z}} X_z$ and $\mathbb{E}[X_z] = (f(z) - f'(z))^2$. Moreover, since $\|f - f'\|_{\mathcal{Z}}^2 \geq \lambda$, by (3) and Definition 3, we have

$$\max_{z \in \mathcal{Z}} X_z \leq \|f - f'\|_{\mathcal{Z}}^2 \cdot \varepsilon^2/(72 \ln(4\mathcal{N}(\mathcal{F}, \varepsilon/72 \cdot \sqrt{\lambda\delta/(|\mathcal{Z}|)})/\delta).$$

Moreover, $\mathbb{E}[X_z^2] \leq (f(z) - f'(z))^4/p_z$. Therefore, by Hölder's inequality,

$$\sum_{z \in \mathcal{Z}} \mathrm{Var}[X_z] \leq \sum_{z \in \mathcal{Z}} \mathbb{E}[X_z^2] \leq \sum_{z \in \mathcal{Z}} (f(z) - f'(z))^2 \cdot \max_{z \in \mathcal{Z}} (f(z) - f'(z))^2/p_z$$

$$\leq \|f - f'\|_{\mathcal{Z}}^4 \cdot \varepsilon^2/(72 \ln(4\mathcal{N}(\mathcal{F}, \varepsilon/72 \cdot \sqrt{\lambda\delta/(|\mathcal{Z}|)})/\delta).$$

Therefore, by Bernstein inequality,

$$\Pr\left[|\|f - f'\|_{\mathcal{Z}}^2 - \|f - f'\|_{\mathcal{Z}'}^2| \geq \varepsilon/4 \cdot \|f - f'\|_{\mathcal{Z}}^2\right]$$

$$= \Pr\left[\left|\sum_{z \in \mathcal{Z}} \mathbb{E}[X_z] - \sum_{z \in \mathcal{Z}} X_z\right| \geq \varepsilon/4 \cdot \|f - f'\|_{\mathcal{Z}}^2\right]$$

$$\leq 2\exp\left(-\frac{\varepsilon^2/16 \cdot \|f - f'\|_{\mathcal{Z}}^4}{2\sum_{z \in \mathcal{Z}} \mathrm{Var}[X_z] + 2\max_{z \in \mathcal{Z}} X_z \cdot \varepsilon/4 \cdot \|f - f'\|_{\mathcal{Z}}^2/3}\right)$$

$$\leq (\delta/4)/\left(\mathcal{N}(\mathcal{F}, \varepsilon/72 \cdot \sqrt{\lambda\delta/(|\mathcal{Z}|)})\right)^2.$$

By union bound, the above inequality implies that with probability at least $1 - \delta/4$, for any $(f, f') \in \mathcal{C}(F, \varepsilon/72 \cdot \sqrt{\lambda\delta/(|\mathcal{Z}|)}) \times \mathcal{C}(F, \varepsilon/72 \cdot \sqrt{\lambda\delta/(|\mathcal{Z}|)})$ with $\|f - f'\|_{\mathcal{Z}}^2 \geq \lambda$,

$$(1 - \varepsilon/4)\|f - f'\|_{\mathcal{Z}}^2 \leq \|f - f'\|_{\mathcal{Z}'}^2 \leq (1 + \varepsilon/4)\|f - f'\|_{\mathcal{Z}'}^2.$$

Now we condition on the event defined above and the event defined in Lemma 3. Consider $f, f' \in \mathcal{F}$ with $\|f - f'\|_{\mathcal{Z}}^2 \geq 2\lambda$. Recall that there exists $(\widehat{f}, \widehat{f'}) \in \mathcal{C}(F, \varepsilon/72 \cdot \sqrt{\lambda\delta/(|\mathcal{Z}|)}) \times \mathcal{C}(F, \varepsilon/72 \cdot \sqrt{\lambda\delta/(|\mathcal{Z}|)})$ such that $\|f - \widehat{f}\|_\infty \leq \sqrt{\lambda/(25|\mathcal{Z}|)}$ and $\|f' - \widehat{f'}\|_\infty \leq \sqrt{\lambda/(25|\mathcal{Z}|)}$. Therefore,

$$\|\widehat{f} - \widehat{f'}\|_{\mathcal{Z}}^2 = \sum_{z \in \mathcal{Z}} (\widehat{f}(z) - \widehat{f'}(z))^2$$

$$= \sum_{z \in \mathcal{Z}} (f(z) - f'(z) + (\widehat{f}(z) - f(z)) + (f'(z) - \widehat{f'}(z)))^2$$

$$\geq \left(\|f - f'\|_{\mathcal{Z}} - \|\widehat{f} - f\|_{\mathcal{Z}} - \|f' - \widehat{f'}\|_{\mathcal{Z}}\right)^2$$

$$\geq \left(\sqrt{2\lambda} - 2\sqrt{\lambda/25}\right)^2 \geq \lambda.$$

Therefore, conditioned on the event defined above, we have

$$(1 - \varepsilon/4)\|\widehat{f} - \widehat{f'}\|_{\mathcal{Z}}^2 \leq \|\widehat{f} - \widehat{f'}\|_{\mathcal{Z}'}^2 \leq (1 + \varepsilon/4)\|\widehat{f} - \widehat{f'}\|_{\mathcal{Z}'}^2.$$

Conditioned on the event defined in Lemma 3 which holds with probability at least $1 - \delta/4$, we have

$$\|f - f'\|_{\mathcal{Z}'}^2 \leq \left(\|\widehat{f} - \widehat{f'}\|_{\mathcal{Z}'} + \|f - \widehat{f}\|_{\mathcal{Z}'} + \|f' - \widehat{f'}\|_{\mathcal{Z}'}\right)^2$$

$$\leq \left(\|\widehat{f} - \widehat{f'}\|_{\mathcal{Z}'} + 2\sqrt{|\mathcal{Z}'|} \cdot \varepsilon/72 \cdot \sqrt{\lambda\delta/(|\mathcal{Z}|)}\right)^2$$

$$\leq \left((1 + \varepsilon/6)\|\widehat{f} - \widehat{f'}\|_{\mathcal{Z}} + 2\sqrt{|\mathcal{Z}'|} \cdot \varepsilon/72 \cdot \sqrt{\lambda\delta/(|\mathcal{Z}|)}\right)^2$$

$$\leq \left((1 + \varepsilon/6)\|f - f\|_{\mathcal{Z}} + 2\sqrt{|\mathcal{Z}'|} \cdot \varepsilon/72 \cdot \sqrt{\lambda\delta/(|\mathcal{Z}|)} + 4\sqrt{|\mathcal{Z}|} \cdot \varepsilon/72 \cdot \sqrt{\lambda\delta/(|\mathcal{Z}|)}\right)^2$$

$$\leq (1 + \varepsilon)\|f - f\|_{\mathcal{Z}}^2,$$

where the last inequality holds since $\|f - f\|_{\mathcal{Z}} \geq \sqrt{\lambda}$.

Similarly,

$$\|f - f'\|_{\mathcal{Z}'}^2 \geq \left(\|\widehat{f} - \widehat{f'}\|_{\mathcal{Z}'} - \|f - \widehat{f}\|_{\mathcal{Z}'} - \|f' - \widehat{f'}\|_{\mathcal{Z}'}\right)^2$$

$$\geq \left(\|\widehat{f} - \widehat{f'}\|_{\mathcal{Z}'} - 2\sqrt{|\mathcal{Z}'|} \cdot \varepsilon/72 \cdot \sqrt{\lambda\delta/(|\mathcal{Z}|)}\right)^2$$

$$\geq \left((1 - \varepsilon/6)\|\widehat{f} - \widehat{f'}\|_{\mathcal{Z}} - 2\sqrt{|\mathcal{Z}'|} \cdot \varepsilon/72 \cdot \sqrt{\lambda\delta/(|\mathcal{Z}|)}\right)^2$$

$$\geq \left((1 - \varepsilon/6)\|f - f\|_{\mathcal{Z}} - 2\sqrt{|\mathcal{Z}'|} \cdot \varepsilon/72 \cdot \sqrt{\lambda\delta/(|\mathcal{Z}|)} - 2\sqrt{|\mathcal{Z}|} \cdot \varepsilon/72 \cdot \sqrt{\lambda\delta/(|\mathcal{Z}|)}\right)^2$$

$$\geq (1 - \varepsilon)\|f - f\|_{\mathcal{Z}}^2.$$

$\square$

Combining Lemma 2, Lemma 3 and Lemma 4 with a union bound, we have the following proposition.

**Proposition 1.** *With probability at least $1 - \delta$, the size of $\mathcal{Z}'$ returned by Algorithm 2 satisfies $|\mathcal{Z}'| \leq 4|\mathcal{Z}|/\delta$, the number of distinct elements in $\mathcal{Z}$ is at most*

$$1728\dim_E(\mathcal{F}, \lambda/|\mathcal{Z}|)\log((H + 1)^2|\mathcal{Z}|/\lambda)\ln(|\mathcal{Z}|)\ln(4\mathcal{N}(\mathcal{F}, \varepsilon/72 \cdot \sqrt{\lambda\delta/(|\mathcal{Z}|)})/\delta)/\varepsilon^2,$$

*and for any $f, f' \in \mathcal{F}$,*

$$(1 - \varepsilon)\|f - f'\|_{\mathcal{Z}}^2 - 2\lambda \leq \|f - f'\|_{\mathcal{Z}'}^2 \leq (1 + \varepsilon)\|f - f'\|_{\mathcal{Z}}^2 + 8|\mathcal{Z}|\lambda/\delta.$$

**Proposition 2.** *For Algorithm 3, suppose $|\mathcal{Z}| \leq KH = T$, the following holds.*

1. *With probability at least $1 - \delta/(16T)$,*

$$w(\underline{\mathcal{F}}, s, a) \leq \widehat{w}(s, a) \leq w(\overline{\mathcal{F}}, s, a)$$

   *where $\underline{\mathcal{F}} = \{f \in \mathcal{F} \mid \|f - \bar{f}\|_{\mathcal{Z}}^2 \leq \beta(\mathcal{F}, \delta)\}$, and $\overline{\mathcal{F}} = \{f \in \mathcal{F} \mid \|f - \bar{f}\|_{\mathcal{Z}}^2 \leq 9\beta(\mathcal{F}, \delta) + 12\}$.*

2. *$\widehat{w}(\cdot, \cdot) \in \mathcal{W}$ for a function set $\mathcal{W}$ with*

$$\log|\mathcal{W}| \leq 6912\dim_E(\mathcal{F}, \delta/(16T^2))\log(16(H + 1)^2T^2/\delta)\ln T \ln(4\mathcal{N}(\mathcal{F}, \delta/(566T))/\delta)$$
$$\cdot \log\left(\mathcal{N}(\mathcal{S} \times \mathcal{A}, 1/(8\sqrt{4T/\delta})) \cdot 4T/\delta\right) + \log(\mathcal{N}(\mathcal{F}, 1/(8\sqrt{4T/\delta})))$$
$$\leq C \cdot \dim_E(\mathcal{F}, \delta/T^3) \cdot \log(H^2T^2/\delta) \cdot \ln T \cdot \ln(\mathcal{N}(\mathcal{F}, \delta/T^2)/\delta)$$
$$\cdot \log\left(\mathcal{N}(\mathcal{S} \times \mathcal{A}, \delta/T)\right) \cdot T/\delta,$$

   *for some absolute constant $C > 0$ if $T$ is sufficiently large.*

*Proof.* For the first part, conditioned on the event defined in Proposition 1, for any $f \in \mathcal{F}$, we have

$$\|f - \bar{f}\|_{\overline{\mathcal{Z}}}^2/2 - 1/2 \leq \|f - \bar{f}\|_{\mathcal{Z}}^2 \leq 3\|f - \bar{f}\|_{\overline{\mathcal{Z}}}^2/2 + 1/2.$$

Therefore, we have

$$\|f - \widehat{f}\|_{\widehat{\mathcal{Z}}}^2 \leq (\|f - \widehat{f}\|_{\overline{\mathcal{Z}}} + \sqrt{4T/\delta}/(8\sqrt{4T/\delta}))^2$$
$$\leq (\|f - \bar{f}\|_{\overline{\mathcal{Z}}} + \sqrt{4T/\delta}/(8\sqrt{4T/\delta}) + \sqrt{4T/\delta}/(8\sqrt{4T/\delta}))^2$$
$$\leq 2\|f - \bar{f}\|_{\overline{\mathcal{Z}}}^2 + 2(\sqrt{4T/\delta}/(8\sqrt{4T/\delta}) + \sqrt{4T/\delta}/(8\sqrt{4T/\delta}))^2 \leq 3\|f - \bar{f}\|_{\mathcal{Z}}^2 + 2$$

and

$$\|f - \widehat{f}\|_{\widehat{\mathcal{Z}}}^2 \geq (\|f - \widehat{f}\|_{\overline{\mathcal{Z}}} - \sqrt{4T/\delta}/(8\sqrt{4T/\delta}))^2$$
$$\geq (\|f - \bar{f}\|_{\overline{\mathcal{Z}}} - \sqrt{4T/\delta}/(8\sqrt{4T/\delta}) - \sqrt{4T/\delta}/(8\sqrt{4T/\delta}))^2$$
$$\geq \|f - \bar{f}\|_{\overline{\mathcal{Z}}}^2/2 - (\sqrt{4T/\delta}/(8\sqrt{4T/\delta}) + \sqrt{4T/\delta}/(8\sqrt{4T/\delta}))^2 \geq \|f - \bar{f}\|_{\mathcal{Z}}^2/3 - 2.$$

Therefore, for any $f \in \underline{\mathcal{F}}$, we have $\|f - \bar{f}\|_{\mathcal{Z}}^2 \leq \beta(\mathcal{F}, \delta)$, which implies $\|f - \widehat{f}\|_{\widehat{\mathcal{Z}}}^2 \leq 3\beta(\mathcal{F}, \delta) + 2$ and thus $f \in \widehat{\mathcal{F}}$. Moreover, for any $f \in \widehat{\mathcal{F}}$, we have $\|f - \widehat{f}\|_{\widehat{\mathcal{Z}}}^2 \leq 3\beta(\mathcal{F}, \delta) + 2$, which implies $\|f - \bar{f}\|_{\mathcal{Z}}^2 \leq 9\beta(\mathcal{F}, \delta) + 12$.

For the second part, note that $\widehat{w}(\cdot, \cdot)$ is uniquely defined by $\widehat{\mathcal{F}}$. When $|\overline{\mathcal{Z}}| \geq 4T/\delta$ or the number of distinct elements in $\overline{\mathcal{Z}}$ exceeds

$$6912\dim_E(\mathcal{F}, \delta/(16T^2))\log(16(H+1)^2T^2/\delta)\ln T\ln(4\mathcal{N}(\mathcal{F}, \delta/(566T))/\delta),$$

we have $|\widehat{\mathcal{Z}}| = 0$ and thus $\widehat{\mathcal{F}} = \mathcal{F}$. Otherwise, $\widehat{\mathcal{F}}$ is defined by $\widehat{f}$ and $\widehat{\mathcal{Z}}$. Since $\widehat{f} \in \mathcal{C}(\mathcal{F}, 1/(8\sqrt{4T/\delta}))$, the total number of distinct $\widehat{f}$ is upper bounded by $\mathcal{N}(\mathcal{F}, 1/(8\sqrt{4T/\delta}))$. Since there are at most

$$6912\dim_E(\mathcal{F}, \delta/(16T^2))\log(16(H+1)^2T^2/\delta)\ln T\ln(4\mathcal{N}(\mathcal{F}, \delta/(566T))/\delta)$$

distinct elements in $\widehat{\mathcal{Z}}$, while each of them belongs to $\mathcal{C}(\mathcal{S} \times \mathcal{A}, 1/(8\sqrt{4T/\delta}))$ and $|\widehat{\mathcal{Z}}| \leq 4T/\delta$, the total number of distinct $\widehat{\mathcal{Z}}$ is upper bounded by

$$\left(\mathcal{N}(\mathcal{S} \times \mathcal{A}, 1/(8\sqrt{4T/\delta})) \cdot 4T/\delta\right)^{6912\dim_E(\mathcal{F}, \delta/(16T^2))\log(16(H+1)^2T^2/\delta)\ln T\ln(4\mathcal{N}(\mathcal{F}, \delta/(566T))/\delta)}.$$

$\square$

## A.2 Analysis of the Algorithm

We are now ready to prove the regret bound of Algorithm 1. The next lemma establishes a bound on the estimate of a single backup.

**Lemma 5** (Single Step Optimization Error). *Consider a fixed $k \in [K]$. Let*

$$\mathcal{Z}^k = \{(s_{h'}^\tau, a_{h'}^\tau)\}_{(\tau, h') \in [k-1] \times [H]}$$

*as defined in Line 5 in Algorithm 1. For any $V : \mathcal{S} \to [0, H]$, define*

$$\mathcal{D}_V^k := \left\{(s_{h'}^\tau, a_{h'}^\tau, r_{h'}^\tau + V(s_{h'+1}^\tau))\right\}_{(\tau, h') \in [k-1] \times [H]}$$

*and*

$$\widehat{f}_V := \arg\min_{f \in \mathcal{F}} \|f\|_{\mathcal{D}_V^k}^2.$$

*For any $V : \mathcal{S} \to [0, H]$ and $\delta \in (0, 1)$, there is an event $\mathcal{E}_{V,\delta}$ which holds with probability at least $1 - \delta$, such that conditioned on $\mathcal{E}_{V,\delta}$, for any $V' : \mathcal{S} \to [0, H]$ with $\|V' - V\|_\infty \leq 1/T$, we have*

$$\left\|\widehat{f}_{V'}(\cdot, \cdot) - r(\cdot, \cdot) - \sum_{s' \in \mathcal{S}} P(s' \mid \cdot, \cdot)V'(s')\right\|_{\mathcal{Z}^k} \leq c' \cdot \left(H\sqrt{\log(2/\delta) + \log \mathcal{N}(\mathcal{F}, 1/T)}\right)$$

*for some absolute constant $c' > 0$.*

*Proof.* In our proof, we consider a fixed $V : \mathcal{S} \to [0, H]$, and define
$$f_V(\cdot, \cdot) := r(\cdot, \cdot) + \sum_{s' \in \mathcal{S}} P(s' \mid \cdot, \cdot)V(s').$$
For any $f \in \mathcal{F}$, we consider $\sum_{(\tau, h) \in [k-1] \times [H]} \xi_h^\tau(f)$ where
$$\xi_h^\tau(f) := 2(f(s_h^\tau, a_h^\tau) - f_V(s_h^\tau, a_h^\tau)) \cdot (f_V(s_h^\tau, a_h^\tau) - r_h^\tau - V(s_{h+1}^\tau)).$$
For any $(\tau, h) \in [k-1] \times [H]$, define $\mathbb{F}_h^\tau$ as the filtration induced by the sequence
$$\{(s_{h'}^t, a_{h'}^t)\}_{(t, h') \in [\tau-1] \times [H]} \cup \{(s_1^\tau, a_1^\tau), (s_2^\tau, a_2^\tau), \dots, (s_h^\tau, a_h^\tau)\}.$$
Then $\mathbb{E}\left[\xi_h^\tau(f) \mid \mathbb{F}_h^\tau\right] = 0$ and
$$|\xi_h^\tau(f)| \le 2(H+1) |f(s_h^\tau, a_h^\tau) - f_V(s_h^\tau, a_h^\tau)|.$$
By Azuma-Hoeffding inequality, we have
$$\Pr\left[\left|\sum_{(\tau, h) \in [k-1] \times [H]} \xi_h^\tau(f)\right| \ge \varepsilon\right] \le 2\exp\left(-\frac{\varepsilon^2}{8(H+1)^2 \|f - f_V\|_{\mathcal{Z}^k}^2}\right).$$
Let
$$\varepsilon = \left(8(H+1)^2 \log\left(\frac{2\mathcal{N}(\mathcal{F}, 1/T)}{\delta}\right) \cdot \|f - f_V\|_{\mathcal{Z}^k}^2\right)^{1/2}$$
$$\le 4(H+1)\|f - f_V\|_{\mathcal{Z}^k} \cdot \sqrt{\log(2/\delta) + \log \mathcal{N}(\mathcal{F}, 1/T)}.$$
We have, with probability at least $1 - \delta$, for all $f \in \mathcal{C}(\mathcal{F}, 1/T)$,
$$\left|\sum_{(\tau, h) \in [k-1] \times [H]} \xi_h^\tau(f)\right| \le 4(H+1)\|f - f_V\|_{\mathcal{Z}^k} \cdot \sqrt{\log(2/\delta) + \log \mathcal{N}(\mathcal{F}, 1/T)}.$$
We define the above event to be $\mathcal{E}_{V,\delta}$, and we condition on this event for the rest of the proof.

For all $f \in \mathcal{F}$, there exists $g \in \mathcal{C}(\mathcal{F}, 1/T)$, such that $\|f - g\|_\infty \le 1/T$, and we have
$$\left|\sum_{(\tau, h) \in [k-1] \times [H]} \xi_h^\tau(f)\right| \le \left|\sum_{(\tau, h) \in [k-1] \times [H]} \xi_h^\tau(g)\right| + 2(H+1)$$
$$\le 4(H+1)\|g - f_V\|_{\mathcal{Z}^k} \cdot \sqrt{\log(2/\delta) + \log \mathcal{N}(\mathcal{F}, 1/T)} + 2(H+1)$$
$$\le 4(H+1)(\|f - f_V\|_{\mathcal{Z}^k} + 1) \cdot \sqrt{\log(2/\delta) + \log \mathcal{N}(\mathcal{F}, 1/T)} + 2(H+1).$$

Consider $V' : \mathcal{S} \to [0, H]$ with $\|V' - V\|_\infty \le 1/T$. We have
$$\|f_{V'} - f_V\|_\infty \le \|V' - V\|_\infty \le 1/T.$$
For any $f \in \mathcal{F}$,
$$\|f\|_{\mathcal{D}_{V'}^k}^2 - \|f_{V'}\|_{\mathcal{D}_{V'}^k}^2 = \|f - f_{V'}\|_{\mathcal{Z}^k}^2 + 2\sum_{(s_{h'}^\tau, a_{h'}^\tau) \in \mathcal{Z}^k} (f(s_{h'}^\tau, a_{h'}^\tau)$$
$$- f_{V'}(s_{h'}^\tau, a_{h'}^\tau)) \cdot (f_{V'}(s_{h'}^\tau, a_{h'}^\tau) - r_{h'}^\tau - V'(s_{h'+1}^\tau)).$$
For the second term, we have,
$$2\sum_{(s_{h'}^\tau, a_{h'}^\tau) \in \mathcal{Z}^k} (f(s_{h'}^\tau, a_{h'}^\tau) - f_{V'}(s_{h'}^\tau, a_{h'}^\tau)) \cdot (f_{V'}(s_{h'}^\tau, a_{h'}^\tau) - r_{h'}^\tau - V'(s_{h'+1}^\tau))$$
$$\ge 2\sum_{(s_{h'}^\tau, a_{h'}^\tau) \in \mathcal{Z}^k} (f(s_{h'}^\tau, a_{h'}^\tau) - f_V(s_{h'}^\tau, a_{h'}^\tau)) \cdot (f_V(s_{h'}^\tau, a_{h'}^\tau) - r_{h'}^\tau - V(s_{h'+1}^\tau))$$
$$- 4(H+1) \cdot \|V' - V\|_\infty \cdot |\mathcal{Z}^k|$$
$$= \sum_{(\tau, h) \in [k-1] \times [H]} \xi_h^\tau(f) - 4(H+1) \cdot \|V' - V\|_\infty \cdot |\mathcal{Z}^k|$$
$$\ge -4(H+1)(\|f - f_V\|_{\mathcal{Z}^k} + 1) \cdot \sqrt{\log(2/\delta) + \log \mathcal{N}(\mathcal{F}, 1/T)} - 2(H+1)$$
$$- 4(H+1) \cdot \|V' - V\|_\infty \cdot |\mathcal{Z}^k|$$
$$\ge -4(H+1)(\|f - f_{V'}\|_{\mathcal{Z}^k} + 2) \cdot \sqrt{\log(2/\delta) + \log \mathcal{N}(\mathcal{F}, 1/T)} - 6(H+1).$$

Recall that $\widehat{f}_{V'} = \arg\min_{f \in \mathcal{F}} \|f\|^2_{\mathcal{D}^k_{V'}}$. We have $\|\widehat{f}_{V'}\|^2_{\mathcal{D}^k_{V'}} - \|f_{V'}\|^2_{\mathcal{D}^k_{V'}} \le 0$, which implies,

$$
\begin{aligned}
0 &\ge \|\widehat{f}_{V'}\|^2_{\mathcal{D}^k_{V'}} - \|f_{V'}\|^2_{\mathcal{D}^k_{V'}} \\
&= \|\widehat{f}_{V'} - f_{V'}\|^2_{\mathcal{Z}^k} \\
&\quad + 2 \sum_{(s^\tau_{h'}, a^\tau_{h'}) \in \mathcal{Z}^k} (\widehat{f}(s^\tau_{h'}, a^\tau_{h'}) - f_{V'}(s^\tau_{h'}, a^\tau_{h'})) \cdot (f_{V'}(s^\tau_{h'}, a^\tau_{h'}) - r^\tau_{h'} - V'(s^\tau_{h'+1})) \\
&\ge \|\widehat{f}_{V'} - f_{V'}\|^2_{\mathcal{Z}^k} \\
&\quad - 4(H+1)(\|\widehat{f}_{V'} - f_{V'}\|_{\mathcal{Z}^k} + 2) \cdot \sqrt{\log(2/\delta) + \log \mathcal{N}(\mathcal{F}, 1/T)} - 6(H+1).
\end{aligned}
$$

Solving the above inequality, we have,

$$
\|\widehat{f}_{V'} - f_{V'}\|_{\mathcal{Z}^k} \le c' \cdot \left( H \cdot \sqrt{\log \delta^{-1} + \log \mathcal{N}(\mathcal{F}, 1/T)} \right)
$$

for an absolute constant $c' > 0$. $\qquad\square$

**Lemma 6** (Confidence Region). *In Algorithm 1, let $\mathcal{F}^k_h$ be a confidence region defined as*

$$
\mathcal{F}^k_h = \left\{ f \in \mathcal{F} \mid \|f - f^k_h\|^2_{\mathcal{Z}^k} \le \beta(\mathcal{F}, \delta) \right\}.
$$

*Then with probability at least $1 - \delta/8$, for all $k, h \in [K] \times [H]$,*

$$
r(\cdot, \cdot) + \sum_{s' \in \mathcal{S}} P(s' \mid \cdot, \cdot) V^k_{h+1}(s') \in \mathcal{F}^k_h,
$$

*provided*

$$
\beta(\mathcal{F}, \delta) \ge c' \cdot \left( H \sqrt{\log(T/\delta) + \log(|\mathcal{W}|) + \log \mathcal{N}(\mathcal{F}, 1/T)} \right)^2
$$

*for some absolute constant $c' > 0$. Here $\mathcal{W}$ is given as in Propostion 2.*

*Proof.* For all $(k, h) \in [K] \times [H]$, the bonus function $b^k_h(\cdot, \cdot) \in \mathcal{W}$. Note that

$$
\mathcal{Q} := \{\min\{f(\cdot, \cdot) + w(\cdot, \cdot), H\} \mid w \in \mathcal{W}, f \in \mathcal{C}(\mathcal{F}, 1/T)\} \cup \{0\}
$$

is a $(1/T)$-cover of

$$
Q^k_{h+1}(\cdot, \cdot) = \begin{cases} \min\{f^k_{h+1}(\cdot, \cdot) + b^k_{h+1}(\cdot, \cdot), H\} & h < H \\ 0 & h = H \end{cases}.
$$

I.e., there exists $q \in \mathcal{Q}$ such that $\|q - Q^k_{h+1}\|_\infty \le 1/T$. This implies

$$
\mathcal{V} := \left\{ \max_{a \in \mathcal{A}} q(\cdot, a) \mid q \in \mathcal{Q} \right\}
$$

is a $(1/T)$-cover of $V^k_{h+1}$ with $\log(|\mathcal{V}|) \le \log|\mathcal{W}| + \log \mathcal{N}(\mathcal{F}, 1/T) + 1$. For each $V \in \mathcal{V}$, let $\mathcal{E}_{V, \delta/(8|\mathcal{V}|T)}$ be the event defined in Lemma 5. By Lemma 5, we have $\Pr\left[\bigcap_{V \in \mathcal{V}} \mathcal{E}_{V, \delta/(8|\mathcal{V}|T)}\right] \ge 1 - \delta/(8T)$. We condition on $\bigcap_{V \in \mathcal{V}} \mathcal{E}_{V, \delta/(8|\mathcal{V}|T)}$ in the rest part of the proof.

Recall that $f^k_h$ is the solution of the optimization problem in Line 8 of Algorithm 1, i.e., $f^k_h = \arg\min_{f \in \mathcal{F}} \|f\|^2_{\mathcal{D}^k_h}$. Let $V \in \mathcal{V}$ such that $\|V - V^k_{h+1}\|_\infty \le 1/T$. Thus, by Lemma 5, we have

$$
\begin{aligned}
&\left\| f^k_h(\cdot, \cdot) - \left( r(\cdot, \cdot) + \sum_{s' \in \mathcal{S}} P(s' \mid \cdot, \cdot) V^k_{h+1}(s') \right) \right\|_{\mathcal{Z}^k} \\
&\le c' \cdot \left( H \sqrt{\log(T/\delta) + \log \mathcal{N}(\mathcal{F}, 1/T) + \log|\mathcal{W}|} \right)
\end{aligned}
$$

for some absolute constant $c'$. Therefore, by a union bound, for all $(k, h) \in [K] \times [H]$, we have $f^k_h(\cdot, \cdot) - \left( r(\cdot, \cdot) + \sum_{s' \in \mathcal{S}} P(s' \mid \cdot, \cdot) V^k_{h+1}(s') \right) \in \mathcal{F}^k_h$ with probability at least $1 - \delta/8$. $\qquad\square$

The above lemma guarantees that, with high probability, $r(\cdot,\cdot) + \sum_{s'\in\mathcal{S}} P(s' \mid \cdot,\cdot)V_{h+1}^k(\cdot,\cdot)$ lies in the confidence region. With this, it is guaranteed that $\{Q_h^k\}_{(h,k)\in[H]\times[K]}$ are all optimistic, with high probability. This is formally presented in the next lemma.

**Lemma 7.** *With probability at least $1 - \delta/4$, for all $(k,h) \in [K] \times [H]$, for all $(s,a) \in \mathcal{S} \times \mathcal{A}$,*

$$Q_h^*(s,a) \le Q_h^k(s,a) \le r(s,a) + \sum_{s'\in\mathcal{S}} P(s'|s,a)V_{h+1}^k(s') + 2b_h^k(s,a).$$

*Proof.* For each $(k,h) \in [K] \times [H]$, define

$$\mathcal{F}_h^k = \left\{ f \in \mathcal{F} \mid \|f - f_h^k\|_{\mathcal{Z}^k}^2 \le \beta(\mathcal{F},\delta) \right\}.$$

Let $\mathcal{E}$ be the event that for all $(k,h) \in [K] \times [H]$, $r(\cdot,\cdot) + \sum_{s'\in\mathcal{S}} P(s' \mid \cdot,\cdot)V_{h+1}^k(s') \in \mathcal{F}_h^k$. By Lemma 6, $\Pr[\mathcal{E}] \ge 1 - \delta/8$. Let $\mathcal{E}'$ be the event that for all $(k,h) \in [K] \times [H]$ and $(s,a) \in \mathcal{S} \times \mathcal{A}$, $b_h^k(s,a) \ge w(\mathcal{F}_h^k, s, a)$. By Proposition 2 and union bound, $\mathcal{E}'$ holds failure probability at most $\delta/8$. In the rest part of the proof we condition on $\mathcal{E}$ and $\mathcal{E}'$.

Note that

$$\max_{f\in\mathcal{F}_h^k} |f(s,a) - f_h^k(s,a)| \le w(\mathcal{F}_h^k, s, a) \le b_h^k(s,a).$$

Since

$$r(\cdot,\cdot) + \sum_{s'\in\mathcal{S}} P(s' \mid \cdot,\cdot)V_{h+1}^k(s') \in \mathcal{F}_h^k,$$

for any $(s,a) \in \mathcal{S} \times \mathcal{A}$ we have

$$\left| r(s,a) + \sum_{s'\in\mathcal{S}} P(s' \mid s,a)V_{h+1}^k(s') - f_h^k(s,a) \right| \le b_h^k(s,a).$$

Hence,

$$Q_h^k(s,a) \le f_h^k(s,a) + b_h^k(s,a) \le r(s,a) + \sum_{s'\in\mathcal{S}} P(s'|s,a)V_{h+1}^k(s') + 2b_h^k(s,a).$$

Now we prove $Q_h^*(s,a) \le Q_h^k(s,a)$ by induction on $h$. When $h = H + 1$, the desired inequality clearly holds. Now we assume $Q_{h+1}^*(\cdot,\cdot) \le Q_{h+1}^k(\cdot,\cdot)$ for some $h \in [H]$. Clearly we have $V_{h+1}^*(\cdot) \le V_{h+1}^k(\cdot)$. Therefore, for all $(s,a) \in \mathcal{S} \times \mathcal{A}$,

$$Q_h^*(s,a) = r(s,a) + \sum_{s'\in\mathcal{S}} P(s'|s,a)V_{h+1}^*(s')$$

$$\le \min\left\{ H, r(s,a) + \sum_{s'\in\mathcal{S}} P(s'|s,a)V_{h+1}^k(s') \right\}$$

$$\le \min\left\{ H, f_h^k(s,a) + b_h^k(s,a) \right\}$$

$$= Q_h^k(s,a).$$

$\square$

The next lemma upper bounds the regret of the algorithm by the sum of $b_h^k(\cdot,\cdot)$.

**Lemma 8.** *With probability at least $1 - \delta/2$,*

$$\mathrm{Reg}(K) \le 2\sum_{k=1}^K \sum_{h=1}^H b_h^k\left(s_h^k, a_h^k\right) + 4H\sqrt{KH \cdot \log(8/\delta)}.$$

*Proof.* In our proof, for any $(k,h) \in [K] \times [H-1]$ define

$$\xi_h^k = \sum_{s'\in\mathcal{S}} P(s' \mid s_h^k, a_h^k)\left(V_{h+1}^k(s') - V_{h+1}^{\pi_k}(s')\right) - \left(V_{h+1}^k(s_{h+1}^k) - V_{h+1}^{\pi_k}(s_{h+1}^k)\right)$$

and define $\mathbb{F}_h^k$ as the filtration induced by the sequence

$$\{(s_{h'}^\tau, a_{h'}^\tau)\}_{(\tau,h')\in[k-1]\times[H]} \cup \{(s_1^k, a_1^k), (s_2^k, a_2^k), \ldots, (s_h^k, a_h^k)\}.$$

Then

$$\mathbb{E}\left[\xi_h^k \mid \mathbb{F}_h^k\right] = 0 \text{ and } |\xi_h^k| \leq 2H.$$

By Azuma-Hoeffding inequality, with probability at least $1 - \delta/4$,

$$\sum_{k=1}^{K}\sum_{h=1}^{H-1} \xi_h^k \leq 4H\sqrt{KH \cdot \log(8/\delta)}.$$

We condition on the above event in the rest of the proof. We also condition on the event defined in Lemma 7 which holds with probability $1 - \delta/4$.

Recall that

$$\text{Reg}(K) = \sum_{k=1}^{K}\left(V_1^*(s_1^k) - V_1^{\pi_k}(s_1^k)\right) \leq \sum_{k=1}^{K} V_1^k(s_1^k) - V_1^{\pi_k}(s_1^k).$$

We have

$$\text{Reg}(K) \leq \sum_{k=1}^{K}\left(r(s_1^k, a_1^k) + \sum_{s'\in\mathcal{S}} P(s' \mid s_1^k, a_1^k)V_2^k(s') + 2b_1^k(s_1^k, a_1^k)\right.$$

$$\left. -r(s_1^k, a_1^k) - \sum_{s'\in\mathcal{S}} P(s' \mid s_1^k, a_1^k)V_2^{\pi_k}(s')\right)$$

$$= \sum_{k=1}^{K}\sum_{s'\in\mathcal{S}} P(s' \mid s_1^k, a_1^k)(V_2^k(s') - V_2^{\pi_k}(s')) + 2b_1^k(s_1^k, a_1^k)$$

$$= \sum_{k=1}^{K} V_2^k(s_2^k) - V_2^{\pi_k}(s_2^k) + \xi_1^k + 2b_1^k(s_1^k, a_1^k)$$

$$\leq \sum_{k=1}^{K} V_3^k(s_3^k) - V_3^{\pi_k}(s_3^k) + \xi_1^k + \xi_2^k + 2b_1^k(s_1^k, a_1^k) + 2b_2^k(s_2^k, a_2^k)$$

$$\leq \sum_{k=1}^{K}\sum_{h=1}^{H-1} \xi_h^k + \sum_{k=1}^{K}\sum_{h=1}^{H} 2b_h^k(s_h^k, a_h^k).$$

Therefore,

$$\text{Reg}(K) \leq 2\sum_{k=1}^{K}\sum_{h=1}^{H} b_h^k(s_h^k, a_h^k) + 4H\sqrt{KH \cdot \log(8/\delta)}.$$

$\square$

It remains to bound $\sum_{k=1}^{K}\sum_{h=1}^{H} b_h^k(s_h^k, a_h^k)$, for which we will exploit fact that $\mathcal{F}$ has bounded eluder dimension.

**Lemma 9.** *With probability at least $1 - \delta/4$, for any $\varepsilon > 0$,*

$$\sum_{k=1}^{K}\sum_{h=1}^{H} \mathbb{I}\left(b_h^k(s_h^k, a_h^k) > \varepsilon\right) \leq \left(\frac{c\beta(\mathcal{F}, \delta)}{\varepsilon^2} + H\right) \cdot \dim_E(\mathcal{F}, \varepsilon)$$

*for some absolute constant $c > 0$. Here $\beta(\mathcal{F}, \delta)$ is as defined in (4).*

*Proof.* Let $\mathcal{E}$ be the event that or all $(k, h) \in [K] \times [H]$,

$$b_h^k(\cdot, \cdot) \leq w(\overline{\mathcal{F}}_h^k, \cdot, \cdot)$$

where
$$\overline{\mathcal{F}}_h^k = \{f \in \mathcal{F} : \|f - f_h^k\|_{\mathcal{Z}^k}^2 \leq 9\beta + 12\}.$$
By Proposition 2, $\mathcal{E}$ holds with probability at least $1 - \delta/4$. In the rest of the proof, we condition on $\mathcal{E}$.

Let $\mathcal{L} = \{(s_h^k, a_h^k) \mid b_h^k(s_h^k, a_h^k) > \varepsilon\}$ with $|\mathcal{L}| = L$. We show that there exists $(s_h^k, a_h^k) \in \mathcal{L}$ such that $(s_h^k, a_h^k)$ is $\varepsilon$-dependent on at least $L/\dim_E(\mathcal{F}, \varepsilon) - H$ disjoint subsequences in $\mathcal{Z}^k \cap \mathcal{L}$. We demonstrate this by using the following procedure. Let $\mathcal{L}_1, \mathcal{L}_2, \ldots, \mathcal{L}_{L/\dim_E(\mathcal{F}, \varepsilon)-1}$ be $L/\dim_E(\mathcal{F}, \varepsilon) - 1$ disjoint subsequences of $\mathcal{L}$ which are initially empty. We consider
$$\{(s_1^k, a_1^k), (s_2^k, a_2^k), \ldots, (s_H^k, a_H^k)\} \cap \mathcal{L}$$
for each $k \in [K]$ sequentially. For each $k \in [K]$, for each $z \in \{(s_1^k, a_1^k), (s_2^k, a_2^k), \ldots, (s_H^k, a_H^k)\} \cap \mathcal{L}$, we find $j \in [L/\dim_E(\mathcal{F}, \varepsilon) - 1]$ such that $z$ is $\varepsilon$-independent of $\mathcal{L}_j$ and then add $z$ into $\mathcal{L}_j$. By the definition of $\varepsilon$-independence, $|\mathcal{L}_j| \leq \dim_E(\mathcal{F}, \varepsilon)$ for all $j$ and thus we will eventually find some $(s_h^k, a_h^k) \in \mathcal{L}$ such that $(s_h^k, a_h^k)$ is $\varepsilon$-dependent on each of $\mathcal{L}_1, \mathcal{L}_2, \ldots, \mathcal{L}_{L/\dim_E(\mathcal{F}, \varepsilon)-1}$. Among $\mathcal{L}_1, \mathcal{L}_2, \ldots, \mathcal{L}_{L/\dim_E(\mathcal{F}, \varepsilon)-1}$, there are at most $H - 1$ of them that contain an element in
$$\{(s_1^k, a_1^k), (s_2^k, a_2^k), \ldots, (s_H^k, a_H^k)\} \cap \mathcal{L},$$
and all other subsequences only contain elements in $\mathcal{Z}^k \cap \mathcal{L}$. Therefore, $(s_h^k, a_h^k)$ is $\varepsilon$-dependent on at least $L/\dim_E(\mathcal{F}, \varepsilon) - H$ disjoint subsequences in $\mathcal{Z}^k \cap \mathcal{L}$.

On the other hand, since $(s_h^k, a_h^k) \in \mathcal{L}$, we have $b_h^k(s_h^k, a_h^k) > \varepsilon$, which implies there exists $f, f' \in \mathcal{F}$ with $\|f - f_h^k\|_{\mathcal{Z}^k}^2 \leq 9\beta + 12$ and $\|f' - f_h^k\|_{\mathcal{Z}^k}^2 \leq 9\beta + 12$ such that $f(z) - f'(z) > \varepsilon$. By triangle inequality, we have $\|f - f'\|_{\mathcal{Z}^k}^2 \leq 36\beta + 48$. On the other hand, since $(s_h^k, a_h^k)$ is $\varepsilon$-dependent on at least $L/\dim_E(\mathcal{F}, \varepsilon) - H$ disjoint subsequences in $\mathcal{Z}^k \cap \mathcal{L}$, we have
$$(L/\dim_E(\mathcal{F}, \varepsilon) - H)\varepsilon^2 \leq \|f - f\|_{\mathcal{Z}^k}^2 \leq 36\beta + 48,$$
which implies
$$L \leq \left(\frac{36\beta + 48}{\varepsilon^2} + H\right) \dim_E(\mathcal{F}, \varepsilon).$$
$\square$

Lastly, we apply the above lemma to bound the overall regret.

**Lemma 10.** *With probability at least $1 - \delta/4$,*
$$\sum_{k=1}^K \sum_1^H b_h^k(s_h^k, a_h^k) \leq 1 + 4H^2 \dim_E(\mathcal{F}, 1/T) + \sqrt{c \cdot \dim_E(\mathcal{F}, 1/T) \cdot T \cdot \beta(\mathcal{F}, \delta)},$$
*for some absolute constant $c > 0$. Here $\beta(\mathcal{F}, \delta)$ is as defined in* (4).

*Proof.* In the proof we condition on the event defined in Lemma 9. We define $w_h^k := b_h^k\left(s_h^k, a_h^k\right)$. Let $w_1 \geq w_2 \geq \ldots \geq w_T$ be a permutation of $\{w_h^k\}_{(k,h)\in[K]\times[H]}$. By the event defined in Lemma 9, for any $w_t \geq 1/T$, we have
$$t \leq \left(\frac{c\beta(\mathcal{F}, \delta)}{w_t^2} + H\right) \dim_E(\mathcal{F}, w_t) \leq \left(\frac{c\beta(\mathcal{F}, \delta)}{w_t^2} + H\right) \dim_E(\mathcal{F}, 1/T),$$
which implies
$$w_t \leq \left(\frac{t}{\dim_E(\mathcal{F}, 1/T)} - H\right)^{-1/2} \cdot \sqrt{c\beta(\mathcal{F}, \delta)}.$$
Moreover, we have $w_t \leq 4H$. Therefore,
$$\sum_{t=1}^T w_t \leq 1 + 4H^2 \dim_E(\mathcal{F}, 1/T) + \sum_{H \dim_E(\mathcal{F}, 1/T) < t \leq T} \left(\frac{t}{\dim_E(\mathcal{F}, 1/T)} - H\right)^{-1/2} \cdot \sqrt{c\beta(\mathcal{F}, \delta)}$$
$$\leq 1 + 4H^2 \dim_E(\mathcal{F}, 1/T) + 2\sqrt{c \cdot \dim_E(\mathcal{F}, 1/T) \cdot T \cdot \beta(\mathcal{F}, \delta)}.$$
$\square$

We are now ready to prove our main theorem.

*Proof of Theorem 1.* By Lemma 8 and Lemma 10, with probability at least $1 - \delta$,

$$\text{Reg}(K) \leq \min\left\{KH, \ \sum_{k=1}^{K}\sum_{h=1}^{H} 2b_h^k\big(s_h^k, a_h^k\big) + 4H\sqrt{KH \cdot \log(8/\delta)}\right\}$$

$$\leq c \cdot \min\left\{KH, \ \left(\dim_E(\mathcal{F}, 1/T) \cdot H^2 \right.\right.$$

$$\left.\left. + \sqrt{\dim_E(\mathcal{F}, 1/T) \cdot T \cdot \beta(\mathcal{F}, \delta).} + H\sqrt{KH \cdot \log \delta^{-1}}\right)\right\}$$

for some absolute constants $c > 0$. Substituting the value of $\beta(\mathcal{F}, \delta)$ completes the proof. $\qquad\square$

## B  Estimating the Sensitivity

In this section, we present a computationally efficient algorithm to estimate the $\lambda$-sensitivity of all state-action pairs in a give set $\mathcal{Z} \subseteq \mathcal{S} \times \mathcal{A}$ with respect to a given function class $\mathcal{F}$. The algorithm is formally described in Algorithm 4.

---

**Algorithm 4** $\mathtt{Estimate}(\mathcal{F}, \mathcal{Z}, \lambda)$

---

1: **Input**: function class $\mathcal{F}$, set of state-action pairs $\mathcal{Z} \subseteq \mathcal{S} \times \mathcal{A}$, accuracy parameter $\lambda$
2: Initialize $\mathsf{sensitivity}^{\text{est}}_{\mathcal{Z},\mathcal{F},\lambda}(z) \leftarrow 0$ for all $z \in \mathcal{Z}$
3: **for** $\alpha \in \{0, 1, \ldots, \log((H+1)^2|\mathcal{Z}|/\lambda) - 1\}$ **do**
4:      Set $N_\alpha \leftarrow |\mathcal{Z}|/\dim_E(\mathcal{F}, (H+1)^2 \cdot 2^{-\alpha-1})$
5:      Initialize $\mathcal{Z}_j^\alpha \leftarrow \{\}$ for each $j \in [N_\alpha]$
6:      **for** $z \in \mathcal{Z}$ **do**
7:          **if** $z$ is dependent on $\mathcal{Z}_j^\alpha$ for all $j \in [N_\alpha]$ **then**
8:              $j^\alpha(z) \leftarrow N_\alpha + 1$
9:          **else**
10:             $j^\alpha(z) \leftarrow \min\{j \in [N_\alpha] \mid z \text{ is independent of } \mathcal{Z}_j^\alpha\}$
11:             Add $z$ into $\mathcal{Z}_j^\alpha$
12:          $\mathsf{sensitivity}^\alpha_{\mathcal{Z},\mathcal{F},\lambda}(z) \leftarrow \frac{2}{j^\alpha(z)}$
13: **for** $z \in \mathcal{Z}$ **do**
14:      $\mathsf{sensitivity}^{\text{est}}_{\mathcal{Z},\mathcal{F},\lambda}(z) \leftarrow \frac{1}{|\mathcal{Z}|} + \sum_{0 \leq \alpha < \log((H+1)^2|\mathcal{Z}|/\lambda)} \mathsf{sensitivity}^\alpha_{\mathcal{Z},\mathcal{F},\lambda}(z)$
15: **return** $\{\mathsf{sensitivity}^{\text{est}}_{\mathcal{Z},\mathcal{F},\lambda}(z)\}_{z \in \mathcal{Z}}$

---

Given a function class $\mathcal{F}$, a set of state-action pairs $\mathcal{Z}$ and an accuracy parameter $\lambda$, Algorithm 4 returns an estimate of the $\lambda$-sensitivity for each $z \in \mathcal{Z}$. With Algorithm 4, we can now implement Algorithm 2 computationally efficiently by replacing (3) in Algorithm 2 with

$$p_z \geq \min\{1, \mathsf{sensitivity}^{\text{est}}_{\mathcal{Z},\mathcal{F},\lambda}(z) \cdot 72\ln(4\mathcal{N}(\mathcal{F}, \varepsilon/72 \cdot \sqrt{\lambda\delta/(|\mathcal{Z}|)})/\delta)/\varepsilon^2\}$$

where for each $z \in \mathcal{Z}$, $\mathsf{sensitivity}^{\text{est}}_{\mathcal{Z},\mathcal{F},\lambda}(z)$ is the estimated $\lambda$-sensitivity returned by Algorithm 4. According to the analysis in Section A.1, to prove the correctness of Algorithm 2 after this modification, it suffices to prove that

$$\mathsf{sensitivity}^{\text{est}}_{\mathcal{Z},\mathcal{F},\lambda}(z) \geq \mathsf{sensitivity}_{\mathcal{Z},\mathcal{F},\lambda}(z)$$

for each $z \in \mathcal{Z}$ and

$$\sum_{z \in \mathcal{Z}} \mathsf{sensitivity}^{\text{est}}_{\mathcal{Z},\mathcal{F},\lambda}(z) \leq 4\dim_E(\mathcal{F}, \lambda/|\mathcal{Z}|)\log((H+1)^2|\mathcal{Z}|/\lambda)\ln|\mathcal{Z}|,$$

which we prove in the remaining part of this section.

**Lemma 11.** *For each $z \in \mathcal{Z}$,* $\mathsf{sensitivity}^{\text{est}}_{\mathcal{Z},\mathcal{F},\lambda}(z) \geq \mathsf{sensitivity}_{\mathcal{Z},\mathcal{F},\lambda}(z)$.

*Proof.* In our proof we consider a fixed $z \in \mathcal{Z}$. Let $f, f' \in F$ be an arbitrary pair of functions such that $\|f - f'\|_{\mathcal{Z}}^2 \geq \lambda$ and

$$\frac{(f(z) - f'(z))^2}{\|f - f'\|_{\mathcal{Z}}^2}$$

is maximized and we define $L(z) = (f(z) - f'(z))^2$ for such $f$ and $f'$. If $L(z) \leq \lambda/|\mathcal{Z}|$, then we have $\mathrm{sensitivity}_{\mathcal{Z},\mathcal{F},\lambda}^{\mathrm{est}}(z) \geq 1/|\mathcal{Z}| \geq \mathrm{sensitivity}_{\mathcal{Z},\mathcal{F},\lambda}(z)$. Otherwise, there exists $0 \leq \alpha < \log((H + 1)^2|\mathcal{Z}|/\lambda)$ such that $L(z) \in ((H + 1)^2 \cdot 2^{-\alpha-1}, (H + 1)^2 \cdot 2^{-\alpha}]$. Since $L(z) > (H + 1)^2 \cdot 2^{-\alpha-1}$ and $z$ is dependent on each of $\mathcal{Z}_1^\alpha, \mathcal{Z}_2^\alpha, \ldots, \mathcal{Z}_{j^\alpha(z)-1}^\alpha$, we have

$$\frac{(f(z) - f'(z))^2}{\|f - f'\|_{\mathcal{Z}}^2} \leq \frac{(H + 1)^2 \cdot 2^{-\alpha}}{(H + 1)^2 \cdot 2^{-\alpha-1}(j^\alpha(z) - 1) + (f(z) - f'(z))^2}$$

$$\leq \frac{2}{j^\alpha(z)} = \mathrm{sensitivity}_{\mathcal{Z},\mathcal{F},\lambda}^\alpha(z) \leq \mathrm{sensitivity}_{\mathcal{Z},\mathcal{F},\lambda}^{\mathrm{est}}(z).$$

$\square$

**Lemma 12.** $\sum_{z \in \mathcal{Z}} \mathrm{sensitivity}_{\mathcal{Z},\mathcal{F},\lambda}^{\mathrm{est}}(z) \leq 4\mathrm{dim}_E(\mathcal{F}, \lambda/|\mathcal{Z}|) \log((H + 1)^2|\mathcal{Z}|/\lambda) \ln|\mathcal{Z}|$.

*Proof.* For each $0 \leq \alpha < \log((H + 1)^2|\mathcal{Z}|/\lambda)$, by the definition of $(H + 1)^2 \cdot 2^{-\alpha-1}$-independence, we have $|\mathcal{Z}_j^\alpha| \leq \mathrm{dim}_E(\mathcal{F}, (H + 1)^2 \cdot 2^{-\alpha-1}) \leq \mathrm{dim}_E(\mathcal{F}, \lambda/|\mathcal{Z}|)$ for each $j \in [N_\alpha]$. Therefore,

$$\sum_{z \in \mathcal{Z}} \mathrm{sensitivity}_{\mathcal{Z},\mathcal{F},\lambda}^\alpha(z) \leq \sum_{z \in \mathcal{Z}} \frac{2}{j^\alpha(z)} = \sum_{j^\alpha(z)=N_\alpha} \frac{2}{j^\alpha(z)} + \sum_{j^\alpha(z)>N_\alpha} \frac{2}{j^\alpha(z)}$$

$$\leq 2\mathrm{dim}_E(\mathcal{F}, \lambda/|\mathcal{Z}|) \ln|\mathcal{Z}| + 2\mathrm{dim}_E(\mathcal{F}, (H + 1)^2 \cdot 2^{-\alpha-1}) \leq 3\mathrm{dim}_E(\mathcal{F}, \lambda/|\mathcal{Z}|) \ln|\mathcal{Z}|.$$

Therefore,

$$\sum_{z \in \mathcal{Z}} \mathrm{sensitivity}_{\mathcal{Z},\mathcal{F},\lambda}^{\mathrm{est}}(z) \leq 3\mathrm{dim}_E(\mathcal{F}, \lambda/|\mathcal{Z}|) \log((H + 1)^2|\mathcal{Z}|/\lambda) \ln|\mathcal{Z}| + 1$$

$$\leq 4\mathrm{dim}_E(\mathcal{F}, \lambda/|\mathcal{Z}|) \log((H + 1)^2|\mathcal{Z}|/\lambda) \ln|\mathcal{Z}|.$$

$\square$

## C  Model Misspecification

In this section, we study the case when there is a misspecification error. Formally, we consider the following assumption.

**Assumption 3.** *There exists a set of functions $\mathcal{F} \subseteq \{f : \mathcal{S} \times \mathcal{A} \to [0, H + 1]\}$ and a real number $\zeta > 0$, such that for any $V : \mathcal{S} \to [0, H]$, there exists $f_V \in \mathcal{F}$ which satisfies*

$$\max_{(s,a)\in\mathcal{S}\times\mathcal{A}} \left| f_V(s, a) - r(s, a) + \sum_{s'\in\mathcal{S}} P(s' \mid s, a)V(s') \right| \leq \zeta.$$

*We call $\zeta$ the* misspecification error.

Our algorithm for the misspecification case is identical the original algorithm except for the change of $\beta(\mathcal{F}, \delta)$. In particular, we change the definition of $\beta(\mathcal{F}, \delta)$ (defined in (4)) as follows.

$$\beta(\mathcal{F}, \delta) = c' \left( H^2 \cdot \log^2\left(\frac{T}{\delta}\right) \cdot \mathrm{dim}_E\left(\mathcal{F}, \frac{\delta}{T^3}\right) \cdot \ln\left(\frac{\mathcal{N}\left(\mathcal{F}, \frac{\delta}{T^2}\right)}{\delta}\right) \right.$$

$$\left. \cdot \log\left(\frac{\mathcal{N}\left(\mathcal{S} \times \mathcal{A}, \frac{\delta}{T}\right) \cdot T}{\delta}\right) + \zeta HT \right) \quad (5)$$

for some absolute constant $c' > 0$. With this, we can now reprove Lemma 5 in the misspecified case.

**Lemma 13** (Misspecified Single Step Optimization Error). *Suppose $\mathcal{F}$ satisfies Assumption 3. Consider a fixed $k \in [K]$. Let*

$$\mathcal{Z}^k = \{(s_{h'}^\tau, a_{h'}^\tau)\}_{(\tau, h') \in [k-1] \times [H]}$$

*as defined in Line 5 in Algorithm 1. For any $V : \mathcal{S} \to [0, H]$, define*

$$\mathcal{D}_V^k := \left\{ \left( s_{h'}^\tau, a_{h'}^\tau, r_{h'}^\tau + V(s_{h'+1}^\tau) \right) \right\}_{(\tau, h') \in [k-1] \times [H]}$$

*and*

$$\widehat{f}_V := \arg\min_{f \in \mathcal{F}} \|f\|_{\mathcal{D}_V^k}^2.$$

*For any $V : \mathcal{S} \to [0, H]$ and $\delta \in (0, 1)$, there is an event $\mathcal{E}_{V, \delta}$ which holds with probability at least $1 - \delta$, such that conditioned on $\mathcal{E}_{V, \delta}$, for any $V' : \mathcal{S} \to [0, H]$ with $\|V' - V\|_\infty \leq 1/T$, we have*

$$\left\| \widehat{f}_{V'}(\cdot, \cdot) - r(\cdot, \cdot) - \sum_{s' \in \mathcal{S}} P(s' \mid \cdot, \cdot) V'(s') \right\|_{\mathcal{Z}^k} \leq c' \cdot \left( H \sqrt{\log(2/\delta) + \log \mathcal{N}(\mathcal{F}, 1/T)} + TH\zeta \right)$$

*for some absolute constant $c' > 0$.*

*Proof.* In our proof, we consider a fixed $V : \mathcal{S} \to [0, H]$, and define

$$f_V(\cdot, \cdot) := r(\cdot, \cdot) + \sum_{s' \in \mathcal{S}} P(s' \mid \cdot, \cdot) V(s').$$

Note that unlike Lemma 5, we may have $f_V \notin \mathcal{F}$. By Assumption 3, we immediately have

$$\min_{f \in \mathcal{F}} \|f - f_V\|_{\mathcal{Z}^k}^2 \leq |\mathcal{Z}^k| \zeta^2 \leq T\zeta^2.$$

For any $f \in \mathcal{F}$, we consider $\sum_{(\tau, h) \in [k-1] \times [H]} \xi_h^\tau(f)$ where

$$\xi_h^\tau(f) := 2(f(s_h^\tau, a_h^\tau) - f_V(s_h^\tau, a_h^\tau)) \cdot (f_V(s_h^\tau, a_h^\tau) - r_h^\tau - V(s_{h+1}^\tau)).$$

Similar to Lemma 5, we still have, with probability at least $1 - \delta$, for all $f \in \mathcal{C}(\mathcal{F}, 1/T)$,

$$\left| \sum_{(\tau, h) \in [k-1] \times [H]} \xi_h^\tau(f) \right| \leq 4(H + 1) \|f - f_V\|_{\mathcal{Z}^k} \cdot \sqrt{\log(2/\delta) + \log \mathcal{N}(\mathcal{F}, 1/T)}.$$

We define the above event to be $\mathcal{E}_{V, \delta}$, and we condition on this event for the rest of the proof. Similarly, we have, for all $f \in \mathcal{F}$,

$$\left| \sum_{(\tau, h) \in [k-1] \times [H]} \xi_h^\tau(f) \right| \leq 4(H + 1)(\|f - f_V\|_{\mathcal{Z}^k} + 1) \cdot \sqrt{\log(2/\delta) + \log \mathcal{N}(\mathcal{F}, 1/T)} + 2(H + 1).$$

Consider $V' : \mathcal{S} \to [0, H]$ with $\|V' - V\|_\infty \leq 1/T$. We still have

$$\|f_{V'} - f_V\|_\infty \leq \|V' - V\|_\infty \leq 1/T.$$

Again by the same argument as in the proof of Lemma 5, we have for any $f \in \mathcal{F}$,

$$\|f\|_{\mathcal{D}_{V'}^k}^2 - \|f_{V'}\|_{\mathcal{D}_{V'}^k}^2 \geq \|f - f_{V'}\|_{\mathcal{Z}^k}^2 - 4(H + 1)(\|f - f_{V'}\|_{\mathcal{Z}^k} + 2)$$
$$\cdot \sqrt{\log(2/\delta) + \log \mathcal{N}(\mathcal{F}, 1/T)} - 6(H + 1).$$

Let $\widetilde{f}_{V'} = \arg\min_{f \in \mathcal{F}} \|f - f_{V'}\|_{\mathcal{Z}^k}^2$. Recall that $\widehat{f}_{V'} = \arg\min_{f \in \mathcal{F}} \|f\|_{\mathcal{D}_{V'}^k}^2$. We have

$$\|\widehat{f}_{V'}\|_{\mathcal{D}_{V'}^k} \leq \|\widetilde{f}_{V'}\|_{\mathcal{D}_{V'}^k} \leq \|f_{V'}\|_{\mathcal{D}_{V'}^k} + \|\widetilde{f}_{V'} - f_{V'}\|_{\mathcal{Z}^k} \leq \|f_{V'}\|_{\mathcal{D}_{V'}^k} + \sqrt{T}\zeta,$$

which implies,

$$
\left( \|\widehat{f}_{V'}\|_{\mathcal{D}_{V'}^k} + \|f_{V'}\|_{\mathcal{D}_{V'}^k} \right) \cdot \sqrt{T}\zeta \geq \|\widehat{f}_{V'}\|_{\mathcal{D}_{V'}^k}^2 - \|f_{V'}\|_{\mathcal{D}_{V'}^k}^2
$$

$$
=\|\widehat{f}_{V'} - f_{V'}\|_{\mathcal{Z}^k}^2
$$
$$
+ 2 \sum_{(s_{h'}^\tau, a_{h'}^\tau) \in \mathcal{Z}^k} (\widehat{f}(s_{h'}^\tau, a_{h'}^\tau) - f_{V'}(s_{h'}^\tau, a_{h'}^\tau)) \cdot (f_{V'}(s_{h'}^\tau, a_{h'}^\tau) - r_{h'}^\tau - V'(s_{h'+1}^\tau))
$$

$$
\geq \|\widehat{f}_{V'} - f_{V'}\|_{\mathcal{Z}^k}^2
$$
$$
- 4(H+1)(\|\widehat{f}_{V'} - f_{V'}\|_{\mathcal{Z}^k} + 2) \cdot \sqrt{\log(2/\delta) + \log \mathcal{N}(\mathcal{F}, 1/T)} - 6(H+1).
$$

Since $\|\widehat{f}_{V'}\|_{\mathcal{D}_{V'}^k} + \|f_{V'}\|_{\mathcal{D}_{V'}^k} \leq 4H\sqrt{T}$, solving the above inequality, we have,

$$
\|\widehat{f}_{V'} - f_{V'}\|_{\mathcal{Z}^k} \leq c' \cdot \sqrt{H^2 \left( \log \delta^{-1} + \log \mathcal{N}(\mathcal{F}, 1/T) \right) + TH\zeta}
$$

for an absolute constant $c' > 0$. $\qquad\square$

Similar to Lemma 6, we have the following lemma.

**Lemma 14** (Misspecified Confidence Region). *Suppose $\mathcal{F}$ satisfies Assumption 3. In Algorithm 1, let $\mathcal{F}_h^k$ be a confidence region defined as*

$$
\mathcal{F}_h^k = \left\{ f \in \mathcal{F} \mid \|f - f_h^k\|_{\mathcal{Z}^k}^2 \leq \beta(\mathcal{F}, \delta) \right\}.
$$

*Then with probability at least $1 - \delta/8$, for all $k, h \in [K] \times [H]$,*

$$
r(\cdot, \cdot) + \sum_{s' \in \mathcal{S}} P(s' \mid \cdot, \cdot) V_{h+1}^k(s') \in \mathcal{F}_h^k,
$$

*provided*
$$
\beta(\mathcal{F}, \delta) \geq c' \cdot \left( H^2 \left( \log(T/\delta) + \log(|\mathcal{W}|) + \log \mathcal{N}(\mathcal{F}, 1/T) \right) + TH\zeta \right)
$$
*for some absolute constant $c' > 0$. Here $\mathcal{W}$ is given as in Proposition 2.*

*Proof.* The proof is nearly identical to that of Lemma 6. $\qquad\square$

Combining Lemma 14 with Lemma 7–10, we obtain the following theorem.

**Theorem 2.** *Under Assumption 3, after interacting with the environment for $T = KH$ steps, with probability at least $1 - \delta$, Algorithm 1 achieves a regret bound of*

$$
\mathrm{Reg}(K) \leq \sqrt{\iota \cdot H^2 \cdot T} + \sqrt{\dim_E(\mathcal{F}, 1/T) \cdot H \cdot \zeta} \cdot T,
$$

*where*

$$
\iota \leq C \cdot \log^2\left(\frac{T}{\delta}\right) \cdot \dim_E^2\left(\mathcal{F}, \frac{\delta}{T^3}\right) \cdot \ln\left(\frac{\mathcal{N}\left(\mathcal{F}, \frac{\delta}{T^2}\right)}{\delta}\right) \cdot \log\left(\frac{\mathcal{N}\left(\mathcal{S} \times \mathcal{A}, \frac{\delta}{T}\right) \cdot T}{\delta}\right)
$$

*for some absolute constants $C > 0$.*