[Reviews · NeurIPS 2020]

Review 1

Summary and Contributions: This paper studies the value based RL and makes progress beyond the tabular and linear setting. The assumptions about the value function class are: (i) it satisfies closedness for all possible value function, and (ii) bounded covering number. The regret is poly in Eluder dimension, log covering number, and has square root in dependence in T. This paper mostly extends [44] (about Eluder dimension) from the bandit case to the RL setting. Although the idea of the algorithm is simply to do regression, the authors construct a very delicate bonus term. The main contribution of this work is to introduce the sensitive sampling [32,20,21] to do universal approximation of the bonus term and significantly reduce the sample size. On the downside, its computation doesn’t seem to be too generally efficient, which might limit the potential impact. Some part of the writing is not clear and can be improved.

Strengths: The understanding beyond the tabular setting and linear value function approximation is limited in theoretical RL field. It's great to see this paper makes progress in this direction and provides a good regret bound. The paper also introduces the sensitive sampling to do universal approximation of the bonus term and at the same time significantly reduce the size of the dataset. Such technique might be useful for the RL community.

Weaknesses: The proposed algorithm doesn’t seem to be computationally efficient in general. It might limit its potential impact.

Correctness: I went into the detail of the proof a little bit. It looks good in general and I have some minor questions (see below).

Clarity: The paper is generally well written and the main text is easy to follow. I think some writing and discussion can be improved to make the paper clearer. The authors didn't mention the assumption 1 (kind of closedness assumption of the function class) in the abstract or in the introduction. I feel it's a very important assumption and it's better to emphasize that at the beginning. It's unclear to me whether the algorithm is really computationally efficient or not. See details below.

Relation to Prior Work: Most of the existing theoretical results (PAC or regret bound) are about the tabular or linear setting. A few algorithms were proposed for more general setting but suffer from the computationally inefficiency. This paper mostly extends [44] (about Eluder dimension) from the bandit case to the RL setting. The idea of the algorithm is simple but the construction of bonus term is quite complicated and technical. Some part of the proof (Lemma 9 and 10) are adapted from the Proposition 3 and Lemma 2 prior work [44]. I think the connection need to be discussed more. It’s unclear to me how the sensitive sampling part differs from the prior work [32,20,21] and I feel more discussions is needed.

Reproducibility: Yes

Additional Feedback: Update: Thanks the authors for addressing my questions. I have read the rebuttal and other reviews, and I would like to keep my score for acceptance. I hope the authors can clarify the closedness assumptions and add more discussions about the sensitive sampling in the later version. ------------------------------------------------------------------------------------------------------------ Nowadays, we have seen many exciting empirical algorithms applying RL to various tasks. However, the theoretical understanding is still limited and most literature is about the tabular setting or linear function approximation setting. The extension beyond these cases are generally hard and under-explored. This paper tackles this problem by proposing a value-based algorithm that extends prior work [44] to the RL setting. The value function class requires two assumptions (i) closedness for all possible value function, and (ii) bounded covering number. The regret bound is poly in Eluder dimension, log covering number, and has square root in dependence in T. Some comments and questions: 1. Due to the definition of Eluder dimension, it's natural to propose the regression-based algorithm. The hard part is how to design a bonus term. It cannot be too small otherwise the optimism part doesn’t go through. It cannot be too large at the same time since the regret bound depends on the magnitude of the bonus term. The main contribution of this work is to introduce the sensitive sampling [32,20,21] to do universal approximation of the bonus term and significantly reduce the sample size. Therefore, width calculated using the smaller dataset differs only in some constant from the value calculated using the whole data (Proposition 2). At the same time, the width function set has bounded complexity related to Eluder dim and log covering number. This lower bound of the width function is used when showing optimism in Lemma 7. The upper bound is applied in Lemma 9, which is helpful when bounding the regret term. The way to handle the relationship between the regret bound and the bonus term in Lemma 9 and 10 are adapted from the Proposition 3 and Lemma 2 prior work [44]. 2. As mentioned above, I think it would be clearer to explicit discussing the closedness assumption in the abstract and the introduction. This assumption is crucial for dynamic programming type of analysis. Although it seems to be just an assumption on the value function class, it actually also implicit makes assumption about the MDP. The difference between the sensitive sampling in this paper and the prior work also need to be discussed more. Moreover, the connection between Lemma 9 and 10 and Proposition 3 and Lemma 2 in [44] also need to be discussed. I think these discussions will be helpful for people to understand the details and digest the proof. 3. In line 52, the authors mention that the proposed algorithm is computationally efficient. However, such argument is unclear to me. In line 255-261, it seems that it requires an oracle to calculate the lambda-sensitivity or to evaluate the independent of the set of state-action pairs w.r.t. a function class F. Therefore, the complexity might scale with |F| or maybe poly in |S|, which is computationally intractable. Can the authors comment on that? 4. I also have a technical question. In line 599, why the conditional expectation is equal to 0?


Review 2

Summary and Contributions: The authors studied an LSVI algorithm in which a general function class is used. They gave performance guarantees which depend on the log covering number and the eluder dimension and has \sqrt{K} dependence. ** I read the author's response and decided to keep my score. Thanks for the clarifications.

Strengths: Generalizes previous algorithms which study LSVI with linear function approximation to a general class of (Q) function approximations. Furthermore, the authors suggest and analyze algorithm 2, which allows them to reduce the computational complexity of the bonus term.

Weaknesses: There are two main weaknesses in my opinion. 1) Assumption 1. The function class needs to be closed w.r.t. every possible value is a harsh assumption. For example, a weaker assumption is closeness w.r.t. to the bellman operator applied on the function class. 2) To calculate D_k one needs to calculate V^k(s_\tau) for all \tau\in[K], i.e., the computational complexity of calculating line 8 is O(K) (and cannot be calculated adaptively using Sherman Morrison formula).

Correctness: I haven't found a problem in the proofs.

Clarity: The text is clearly written.

Relation to Prior Work: Relation to previous works are clearly discussed.

Reproducibility: Yes

Additional Feedback: *) Computational complexity. The algorithm needs to calculate the current value function V^k on the entire historical data. This makes the algorithm run in O(T). Think it is something that is not stressed enough. *) Proof of Lemma 9. In the proof you bound the bonus using properties of the function class F (i.e, its eluder dimension). How the bonus is related to the properties of the function class? I think an explanation would make this proof much more readable. *) Is there a reason not to use the doubling trick to reduce the computational complexity? Is there something that prevents it? I think the authors should comment about it in the paper.


Review 3

Summary and Contributions: The paper extends the low-rank idea recently introduced by Jin et al '20 to more general, i.e., non-linear predictors, showing 1) connection to the Eluder dimension and 2) reducing the complexity of the bonus to reduce the size of the cover by selecting the most relevant data. ======================= After discussion with the other reviewers and area chair, I raised my score with the understanding that the authors will better clarify the relations with the literature, the assumptions, and other concerns other reviewers may have

Strengths: Point 1 and 2 described in the summary above are (indeed very good) strengths of this submission.

Weaknesses: I report these under weaknesses but it's more like observations - Assumption: This is an extension of the low-rank setting in that it maps any value function to a prescribed functional space (which is the key property used in the linear MDP setting of Jin '20, regardless of the decomposition into a low-rank MDP). This should be better emphasized by the author(s): the reason why they are able to proceed in the analysis using traditional methods is because they can add bonuses and still be able to represent the value function as the Bellman operator automatically maps everything back to the prescribed functional space. The assumption is certainly strong (as is the linear MDP assumption in the linear case); while some assumptions have to be made, it should be motivated better - The paper operates in a very general setting in terms of predictors, where we don't normally have confidence intervals available, so the authors just assume that. Owning to the strong assumption (see bullet points above), the authors create a bonus equal to the confidence intervals. This effectively solves most of the exploration problem by assumption (the estimation part is often difficult, and the construction of the bonus might be very hard). Not much can be said in such a general setting, but all this bypasses most of the difficulties.

Correctness: Yes

Clarity: Yes

Relation to Prior Work: The works makes two claims which I think sends the wrong message - please either elaborate on these by including a proof in the manuscript or don't include them unless absolutely certain. 1) around line 272. "By a more refined analysis specialized to the tabular setting, the regret bound of our algorithm can be improved using techniques in [6]" I'm not sure this is the case. The work [6] uses an exploration bonus adapted to the variance of the system dynamics. Works prior to [6] have been unable to obtain the same regret bound as [6] without this. This suggest that your algorithm for tabular setting might need to be modified, and the bound does not improve merely by improving the analysis as you claim - Line 284. "Finally, we would like to stress that our algorithm and analysis tackle the case that F is a general function class which contains the linear case studied in [27, 57, 62] as a special case." Reference [62] does not make the assumption that any value function comes out in the prescribed functional space (linear in their case), so I think the two are incomparable works in terms of assumptions.

Reproducibility: Yes

Additional Feedback:


Review 4

Summary and Contributions: The paper studies exploration-exploitation dilemma in episodic reinforcement learning with general function approximation. The proposed algorithm, F-LSVI, is an optimistic modification of least square value iteration with general function class F. Under some conditions on F, the algorithm enjoys a sublunar regret that depends on the Eluder dimension and the log-coving number of F.

Strengths: It is a new step towards sample-efficient model-free reinforcement learning with general function approximation. It certainly provides some new intuition about how to achieve sample efficiency in challenging setting beyond tabular representation or linear MDP. The proposed algorithm enjoys nice theoretical guarantees. The sensitivity sampling looks interesting approach and it is a key element of the algorithm to control its regret.

Weaknesses: Assumption 1 seems to induce implicit condition on the model of the environment. The choice the function class F would be intimately related to the structure of the environment at hand, for example in linear MDP, the function class would be the linear function with the same features that defines the low rank structure of the transition model. So I am not sure about the author ’s claim that they don’t make any assumption on the model of the environment. I have some concerns about the computational efficiency. Beyond the linear case (which is already addressed by prior works), the bonus function which is defined by the width function doesn’t look something that we can’t estimate easily and naive approach would scale with the covering number of F. The same holds for determining whether z is independent of a given set with respect to F. The definition of beta in the confidence region looks very pessimistic. I think it would lead to over-exploration in practice. For example, the covering number of the class of Lipschitz functions is doubly exponential in the dimension which lead to value of beta exponential in the dimension. It would be interesting to have some empirical results on simple environment to highlight advantages and disadvantages of some components of the algorithm (the importance of sensitivity sampling for example)

Correctness: I went through the poof but not in detail so I can’t really judge the correctness of every steps of the analysis but it looks overall sounded. I encourage the author to improve the clarity of the appendix, there are many typos (eg: \bar{Z} in line 578 is not defined before, the inequality after line 532 should be the inverse..). In addition, in the proof of lemma 1 as well as in the algorithm 4 in appendix, It is not clear at all how Z^\alpha_j are constructed, Z^\alpha_j are set to {} but it is not mentioned how they are updating. Could you please clarify this?

Clarity: The paper is well written.

Relation to Prior Work: Discussion with the recent closely related work (Ayoub et al 2020) is missing. In fact, beyond model-free and model-based differences, (Ayoub et al 2020) study the setting of general function approximation and I think their assumption is very related to the one made in this paper and prove similar regret bound in their theorem 1. Model-Based Reinforcement Learning with Value-Targeted Regression, 2020 Alex Ayoub , Zeyu Jia , Csaba Szepesvari , Mengdi Wang and Lin F. Yang

Reproducibility: Yes

Additional Feedback:

[Author Response · NeurIPS 2020]

We thank all the reviewers for their valuable feedback. We first address some common concerns.

**Assumption 1.** We will emphasize this assumption at the beginning of the paper. We will mention that although this
assumption does not make explicit assumption about the model, it makes implicit assumption about the MDP.

**Computational efficiency.** We have provided an algorithm in Appendix C to compute the $\lambda$-sensitivity. This algorithm
requires only an oracle to test whether a given state-action pair is $\varepsilon$-independent with a sequence of state-action pairs,
which is again equivalent to evaluating the width function (defined in Line 140). To evaluate the width function, it
suffices to have access to a regression oracle by invoking known reductions in [1] and [2], and having access to a
regression oracle is indeed a weak assumption in practice. We will add more discussion on the computational efficiency.
[1] Practical Contextual Bandits with Regression Oracles       [2] Active Learning for Cost-Sensitive Classification

—— **To Reviewer #1** ——

**The difference between the sensitive sampling in this paper and the prior work.** Sensitivity sampling was proposed
and applied in a different context (e.g., in [20, 21, 32] for clustering) to compress datasets. Our definition of sensitivity
is similar to previous results, and the main technical novelty here is that we can show the sum of the sensitivity can be
upper bounded in terms of the eluder dimension of the function class (Lemma 1). Such a result is crucial for obtaining
an upper bound on the complexity of the bonus function. We will add more comparison in the next version.

**How sensitivity sampling helps address the problem.** To account for the dependency structure in the data sequence,
we need to construct a bonus function with bounded complexity, and thus we subsample the dataset to reduce its size.
As mentioned in Line 232-233, sensitivity measures the importance of each data point $z$ in a dataset. In our analysis
(Proposition 1), we show that by importance sampling according to the sensitivity, the subsampled dataset has bounded
size while the confidence region is approximately preserved, and thus the bonus function has bounded complexity.

**The connection between Lemma 9 and 10 and Proposition 3 and Lemma 2 in [44].** Lemma 9 and 10 are indeed
adapted from Proposition 3 and Lemma 2 in [44], and the main difference is that our confidence region is defined using
the subsampled dataset. We have discussed this in Line 331-336, and we will make this clearer in the next version.

**Line 599.** There is a typo here in the definition of $\mathbb{F}_h^\tau$, which should also includes $(s_h^\tau, a_h^\tau)$. Conditioned on $\mathbb{F}_h^\tau$ ($(s_h^\tau, a_h^\tau)$
is fixed and $s_{h+1}^\tau$ is random), $\mathbb{E}[V(s_{h+1}^\tau)] = \sum_{s' \in \mathcal{S}} P(s'|s_h^\tau, a_h^\tau)V(s')$, and thus the conditional expectation is 0.

—— **To Reviewer #2** ——

**Computational complexity / doubling trick.** The running time of the current algorithm is polynomial in $T$, which we
will emphasize more in the next version. We do believe the running time can be further reduced by using the doubling
trick or online sampling algorithms, and we leave it as a future work to further optimize the running time.

**The bonus is related to the properties of the function class.** Lemma 9 is adapted from prior work [44] (to handle
confidence regions defined using the subsampled dataset), and is analogous to the elliptical potential lemma in the linear
case. Note that for the linear case, the summation of the bonus function can be upper bounded by $\widetilde{O}(d)$, and the feature
dimension $d$ is also a property of the function class.

—— **To Reviewer #3** ——

**Assumption.** We agree with the reviewer that our assumption allows us to add bonuses and still be able to represent the
value function, and we will make this more explicit. However, since we work with *general value function* approximation
in this paper, we do need to make some assumptions to make the problem tractable. We would like to remind the
reviewer that even for the case of linear functions, assumptions weaker than linear MDP either result in computationally
inefficient algorithms (as in [62]) or require the transition to be (nearly) deterministic (as in [18, 19]).

**Confidence intervals.** Note that for the class of linear functions, the width function defined in Line 140 recovers the
usual confidence interval ($\|\phi\|_{\Sigma^{-1}}$) for linear functions. It is not clear to us why the reviewer believes that we "just
assume that the confidence intervals are available". Moreover, our definition of the bonus function requires non-trivial
effort by extending techniques from sensitivity sampling to make sure the complexity of the bonus function is low. We
believe this method itself could be of interest to the machine learning community in general.

**Two claims that sends the wrong message.** (1) By "by a more refined analysis specialized to the tabular setting, ...",
we mean the sample complexity can be improved from $\widetilde{O}(\sqrt{|\mathcal{S}|^3|\mathcal{A}|^3 H^2 T})$ to $\widetilde{O}(\sqrt{|\mathcal{S}|^2|\mathcal{A}|^2 H^2 T})$. We will remove
that sentence from Remark 1 in the next version. (2) We will remove [62] from that list to avoid possible confusion.

—— **To Reviewer #4** ——

**Typos.** Thanks for pointing out the typos, we will polish the paper and fix these typos in the next version.

**Constructing $\mathcal{Z}_j^\alpha$.** For each data $z$, we should also add $z$ into $\mathcal{Z}_{j(z)}^\alpha$ if $j(z) \le N_\alpha$. Sorry for missing this step.

**The confidence region looks very pessimistic.** Our confidence interval recovers the usual confidence interval ($\|\phi\|_{\Sigma^{-1}}$)
for linear functions and thus correctly balances exploration and exploitation in that case. Also, as we show in Lemma
10, the summation of the bonus function can be upper bounded in terms of the eluder dimension of the function class,
and thus provides a finite regret bound. We will make this clear in the final version.

**Empirical results.** We will consider adding empirical results in the next version. Thanks for the suggestion.

**Comparison with Ayoub et al.** For linear functions, our assumption is equivalent to the linear MDP assumption, where
the assumption in Ayoub et al. assumes that the true model is a linear combination of some known models. Therefore,
these two assumptions are already incomparable for linear functions, and we will make this clear in the next version.

[Meta-Review · NeurIPS 2020]

The problem of exploration in RL with function approximation is very important and any advancement on the topic is of interest for the community. The reviewers all agreed about the algorithmic and technical contribution of the paper, in particular the introduction of sensitive sampling and its analysis in the regret proof. This convinced us that the paper deserves acceptance. Nonetheless, I also encourage the authors to improve the current submission. As pointed out by R3, the assumptions used in the paper are quite strong and they may somehow limit the generality of the results. The authors should stress the potential limitations coming from the assumptions and better contrasted it with the related literature. In particular, using the Eluder dimension as a measure of complexity is definitely interesting but, as of today, it lacks of interpretability. In fact, we have very few families of MDPs for which a meaningful bound for the Eluder dimension is available (i.e., linear and GLM). The authors should point this out and possibly clarify whether it is possibly to obtained good bounds for other classes of problems.